# LLM-Guided Search for Deletion-Correcting Codes

## Abstract

Finding deletion-correcting codes of maximum size has been an open problem for over 70 years, even for a single deletion. In this paper, we propose a novel approach for constructing deletion-correcting codes. A code is a set of sequences satisfying certain constraints, and we construct it by greedily adding the highest-priority sequence according to a priority function. To find good priority functions, we leverage FunSearch, a large language model (LLM)-guided evolutionary search proposed by Romera et al., 2024. FunSearch iteratively generates, evaluates, and refines priority functions to construct large deletion-correcting codes. For a single deletion, our evolutionary search finds functions that construct codes which match known maximum sizes, reach the size of the largest (conjectured optimal) Varshamov-Tenengolts codes where the maximum is unknown, and independently rediscover them in equivalent form. For two deletions, we find functions that construct codes with new best-known sizes for code lengths $n = 12, 13$, and 16, establishing improved lower bounds. These results demonstrate the potential of LLM-guided search for information theory and code design and represent the first application of such methods for constructing error-correcting codes.

## 1 Introduction

Error-correcting codes enable reliable communication and data recovery from storage media (such as HDDs and SSDs), even in the presence of errors and defects. In a typical coding scheme, an encoder maps information to a codeword, which is corrupted by errors during transmission, and a decoder attempts to recover the original message. While substitutions and erasures are well understood with optimal encoding and decoding algorithms approaching known theoretical limits, deletions are significantly more challenging. Deletions shift subsequent symbols, disrupting the memoryless property typically assumed in coding theory.

Correcting deletions is of theoretical and practical interest. In theoretical computer science, problems related to deletion errors include determining whether the edit distance between two strings can be computed in strongly sub-quadratic time (Backurs & Indyk, 2015). Deletion errors are practically relevant in cryptography (Bartusek & Khurana, 2023), multiple sequence alignment in computational biology (Carrillo & Lipman, 1988), document exchange (Cheng et al., 2018), traditional storage technologies such as racetrack memories (Bläsing et al., 2020; Parkin et al., 2008) and bit-patterned magnetic recording (Albrecht et al., 2015), as well as emerging technologies such as DNA data storage (Gimpel et al., 2024).

For a fixed number of correctable errors, better codes have larger code sizes. Despite significant effort, determining the maximum code size for a fixed number of adversarial deletions has proven difficult using traditional hand-crafted, human-driven approaches to information theory. A class of codes known as Varshamov-Tenengolts (VT) codes (Varshamov & Tenengolts, 1965) achieves the maximum code size for correcting a single deletion as the code length approaches infinity (Levenshtein, 1966). However, for finite code lengths, the gap to the best-known upper bound is large even at moderate code lengths (Kulkarni & Kiyavash, 2013). Although VT codes are conjectured to be maximum-size for all code lengths and a single deletion, their optimality has only been proven for lengths up to 11 (Butenko et al., 2002; Nakasho et al., 2023; Sloane, 2002).

In this paper, we propose a novel approach for constructing error-correcting codes using large language models (LLMs) and evolutionary search. While our framework is general, we focus on binary

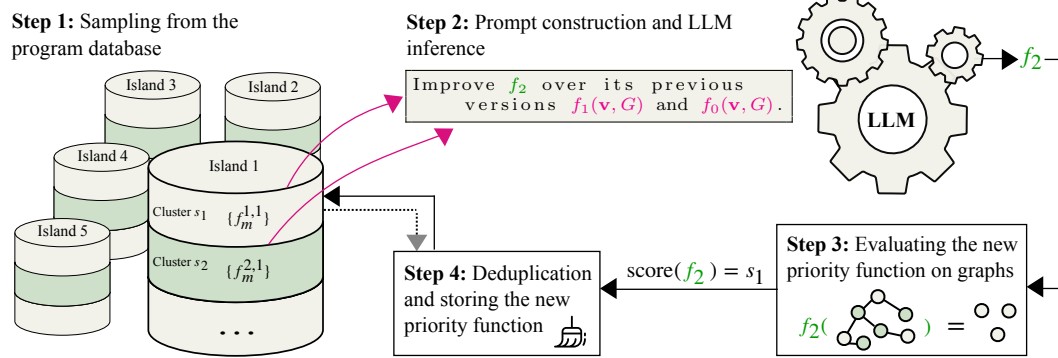

Figure 1: FunSearch for finding deletion-correcting codes iteratively refines a priority function through evolutionary search guided by a pretrained LLM. In each iteration, a few-shot prompt is constructed by sampling from the program database. The LLM generates a new priority function, which is evaluated by greedily constructing deletion-correcting codes for different code lengths and numbers of deletions. If executable and not a duplicate, the function is added to the database.

codes that correct a fixed number of adversarial deletions, as many fundamental questions remain open in this setting (Sloane, 2002). We find explicit algorithms that construct deletion-correcting codes by assigning priorities to sequences. These algorithms build the code greedily by iteratively adding the highest-priority sequence while ensuring that deletion-correcting constraints are satisfied.

LLMs are successful for challenging tasks such as mathematical reasoning and coding (Chen et al., 2021; Cobbe et al., 2021; Lewkowycz et al., 2022; Li et al., 2022), but are often limited to their training data and existing knowledge (Bender et al., 2021; Mahowald et al., 2024). Recently, Romera-Paredes et al. (2024) showed that combining LLMs with evolutionary search and an external evaluator can overcome this limitation for problems that are difficult to solve but easy to evaluate. Their method, FunSearch (Function Space Search), represents combinatorial problems as python code and searches for algorithmic solutions, improving on best-known results for problems such as the cap set problem and the online bin-packing problem.

We adapt FunSearch (Romera-Paredes et al., 2024) to find large deletion-correcting codes. Our main contributions are:

- We propose an LLM-guided evolutionary search to find deletion-correcting codes based on FunSearch.

- Our search discovers functions that construct previously unknown maximum-size codes for a single deletion and small code lengths ($n \leq 11$), and match the size of the conjectured-optimal VT codes for larger code lengths (verified up to $n = 25$), including one that independently rediscovers them. For two deletions, we find improved lower bounds for code lengths $n = 12, 13$ and $16$.

- We provide an efficient, parallel implementation of the LLM-guided evolutionary search and release our code alongside the paper to facilitate future research.

Our results demonstrate the potential of LLM-guided search for information and coding theory. However, our current approach does not scale well to long codes, a limitation we discuss in more detail later.

## 2 RELATED WORK

We review related work on LLM-guided search and deletion-correcting codes.

### 2.1 RELATED WORK ON LLM-GUIDED SEARCH

As mentioned, our work builds on FunSearch (Romera-Paredes et al., 2024). Other approaches also integrate LLMs in evolutionary search. Lehman et al. (2023) first demonstrate a synergy between

LLMs and evolutionary search, using the LLM as an intelligent mutator for automatic data generation. Other applications of LLM-guided search are in machine learning (Chen et al., 2023; Fernando et al., 2024; Hazra et al., 2024; Lee et al., 2025; Lu et al., 2024; Ma et al., 2023; Nasir et al., 2024; Shojaee et al., 2024; Yang et al., 2023; Zheng et al., 2023), black-box optimization (Aglietti et al., 2024; Brahmachary et al., 2025; Lange et al., 2024), system-level research (Cheng et al., 2025), and automatic heuristic design.

The most relevant application to finding deletion-correcting codes is automatic heuristic design for combinatorial problems. Liu et al. (2024) propose EoH, which improves performance and sample efficiency over FunSearch by evolving both natural language and algorithmic components. Ye et al. (2024) introduce ReEvo, which incorporates reflection into the search by prompting the LLM to compare previously generated solutions. ReEvo improves sample efficiency over FunSearch at the cost of increased inference per iteration. Dat et al. (2024) propose two diversity metrics and find that FunSearch and ReEvo stagnate in local optima due to low diversity, while EoH trades off diversity for performance. To address the tradeoff, they tune function parameters via harmony search (Shi et al., 2012).

None of the methods building on FunSearch (Chen et al., 2024; Dat et al., 2024; Liu et al., 2024; Ye et al., 2024; Zheng et al., 2025) outperform the results discovered by FunSearch on large-scale instances of the cap set problem. This suggests that scaling LLM-based evolutionary search in a distributed system is important for solving certain combinatorial problems. We provide a suitable, scalable implementation.

Recent work, AlphaEvolve (Novikov et al., 2025), generalizes FunSearch by evolving entire codebases across multiple files rather than individual functions. Georgiev et al. (2025) further combine AlphaEvolve with proof-assistant and reasoning systems to enable automated proof generation and large-scale mathematical discovery across combinatorics, geometry, and number theory.

## 2.2 RELATED WORK ON DELETION-CORRECTING CODES

Levenshtein (1966) proves that VT codes (Varshamov & Tenengolts, 1965) are asymptotically optimal for correcting a single deletion and proposes a linear-time decoding algorithm. VT codes are also conjectured to be largest for finite code lengths $n$, but this has only been proven for $n \leq 11$ (for $n \leq 8$, Sloane (2002); for $n \leq 10$, Butenko et al. (2002); for $n \leq 11$, Nakasho et al. (2023)).

Levenshtein (2002) derives non-asymptotic upper and lower bounds for single-deletion-correcting codes. Later work (Cullina & Kiyavash, 2016; Fazeli et al., 2015; Kulkarni & Kiyavash, 2013) refines his upper bound by formulating the problem as a linear program and considering its dual relaxation. The optimal solution to the relaxation equals the relaxation of the original problem and provides an upper bound on the maximum code size. However, exhaustive search by Kulkarni & Kiyavash (2013) for short code lengths shows a gap between the best relaxed solution and the largest VT codes.

Regarding known constructions for multiple deletions, Helberg & Ferreira (2002) extend VT codes and propose a first explicit construction, but the resulting code sizes are suboptimal for longer lengths. Swart & Ferreira (2003) find larger code sizes for two deletions and code lengths $n \leq 12$ by using a run-length representation of sequences in a greedy search over $5 \times 10^4$ random permutations. Similarly, Landjev & Haralambiev (2007) use heuristics and search in the space of all binary sequences to construct deletion-correcting codes for code lengths $n \leq 30$ and deletions $s = 2, 3, 4, 5$.

## 3 PROBLEM STATEMENT

We consider the problem of constructing large $n$-bit, $s$-deletion-correcting codes.

A deletion-correcting code is a set of sequences such that, even if an adversary deletes $s$ bits from a sequence, the original sequence can still be uniquely recovered. Unique recovery is not possible if two sequences share a common subsequence of length $n - s$. A subsequence is any sequence of length $n - s$ obtained by deleting $s$ bits from the original sequence while preserving the order of the remaining bits. Thus, an $n$-bit, $s$-deletion-correcting code is a set $\mathcal{C} \subseteq \{0, 1\}^n$ such that the sets of length $(n - s)$ subsequences obtained from any two distinct sequences $\mathbf{c}, \mathbf{c}' \in \mathcal{C}$ are disjoint.

```
"""
Finds large independent set in graph G where vertices are binary strings of length n.
Vertices in G are connected if they share a subsequence of length at least n − s.
Improve f₁ over its previous versions below.
Keep the code short and comment for easy understanding.
"""
import numpy as np
import networkx as nx

def f₀(v, G):
    """Returns the priority with which we want to add vertex v."""
    return 0.0

def f₁(v, G):
    """Improved version of f₀"""
```

Figure 2: Initial prompt with function $f_0$ that initializes all islands.

The problem of constructing large $n$-bit, $s$-deletion-correcting codes can be reduced to finding an independent set $\mathcal{I}$ in a graph $G$ defined as follows. Let $G$ be an undirected graph where each vertex is one of the $2^n$ binary sequences of length $n$, and we have an edge between two vertices if and only if the binary sequences they represent share a common subsequence of length at least $n - s$. An independent set in the graph $G$ is a subset of vertices $\mathcal{I}$ such that no two vertices are connected by an edge. An $n$-bit, $s$-deletion-correcting code is an independent set in the graph $G$.

To construct deletion-correcting codes, we greedily build independent sets $\mathcal{I}$ in the graph $G$ by iteratively adding vertices $\mathbf{v}$ with the highest priority to an initially empty set and removing their neighbors. Let $f(\mathbf{v}, G)$ be a priority function that assigns a real-valued priority to each vertex $\mathbf{v}$ in the graph $G$. At each step, we select the vertex with the highest priority $f(\mathbf{v}, G)$, add it to the independent set $\mathcal{I}$, and remove the vertex and its neighbors from $G$. If two or more vertices have the same priority, we break the tie by selecting the lexicographically smallest vertex (with $0$ considered smaller than $1$). The size of the resulting independent set $\mathcal{I}$ depends on the choice of the priority function $f$, which determines which vertices are added.

In this formulation, constructing large $n$-bit, $s$-deletion-correcting codes reduces to designing a priority function $f$ that maximizes the independent set size $\mathcal{I}$ in the graph $G$.

## 4 METHOD

We adapt FunSearch (Romera-Paredes et al., 2024) with a deduplication step to optimize the priority function $f$ to construct large deletion-correcting codes. FunSearch works iteratively. In each iteration, we sample existing priority functions as examples, prompt an LLM with these examples to generate a new priority function and evaluate the generated priority function by constructing codes as described in Section 3. We explain each step below.

### 4.1 SAMPLING FROM THE PROGRAM DATABASE.

We organize priority functions in a program database divided into independent sub-databases called islands. We sample few-shot examples from a single island $j$ and store new functions back on the same island, allowing the islands to explore independently.

Within each island, we group functions into clusters. Two functions belong to the same cluster $i$ if they achieve the same independent set sizes on all evaluation inputs. All functions in a cluster therefore share the same performance, and we assign this shared performance a single value $\text{score}_i$.

To sample a priority function, we first select an island $j$ uniformly at random. We then sample a cluster $i$ from island $j$ with probability

$$p_i = \frac{e^{\text{score}_i / T_j}}{\sum_{i'} e^{\text{score}_{i'} / T_j}}, \quad \text{where } T_j = T\left(1 - \frac{n_j \bmod P}{P}\right).$$

The temperature $T_j$ balances exploration and exploitation on island $j$. It depends on an initial temperature $T$, the number of stored functions $n_j$, and a sampling period $P$. As $n_j$ increases, the tem-

perature decreases, shifting from uniform sampling (exploration) to favoring high-scoring clusters (exploitation). The temperature resets every $P$ functions to periodically reintroduce exploration.

Finally, from the sampled cluster $i$, we sample a priority function $f$, favoring shorter functions based on their length relative to the cluster's minimum and maximum function lengths. This follows from Kolmogorov complexity (Kolmogorov, 1965; Li et al., 2008), as shorter functions often have lower computational complexity and evaluate faster, though this is not always true in practice.

### 4.2 Prompt construction and LLM inference.

We construct a few-shot prompt by sampling twice from the database as described above to obtain two priority functions. Sampling is done without replacement for diverse examples. The priority functions are sorted by their cluster scores, with the lower-scoring function first and the higher-scoring function second as a target for improvement. The prompt is framed as a code completion task, ending with the header of a new priority function for the LLM to complete. The initial prompt is shown in Figure 2.

We use StarCoder2-15B (Lozhkov et al., 2024) to generate new priority functions, an open-access model with 15 billion parameters trained on The Stack v2 dataset (775B tokens from 600+ programming languages) and additional tokens from sources like pull requests, issues, Jupyter notebooks, and StackOverflow, totaling 913B tokens.

### 4.3 Evaluating the new priority function on graphs.

For each evaluation input that consist of a code length $n$ and a deletion parameter $s$, we use the new priority function to construct an independent set $\mathcal{I}$ in graph $G$, i.e., a deletion-correcting code. If the function is not executable (e.g., due to syntax errors), we discard it.

If executable, we assign the priority function a score. The score is the independent set size it achieves on the longest code length, which we found to outperform alternatives such as averaging sizes across all code lengths or using weighted combinations (see Appendix E).

### 4.4 Deduplication and storing the new priority function.

We store the new priority function on island $j$, the same island from which we sampled the few-shot examples. We compare the independent set sizes it achieves across all evaluation inputs to existing clusters on island $j$. If no cluster exists with functions achieving the same sizes, we create a new cluster and assign it the function's score.

If a matching cluster exists, we check for duplicates before adding the function. Two functions are duplicates if they assign the same priorities to all sequences. If the function is not a duplicate, we add it to the cluster. If it is a duplicate, we discard it. Our deduplication step allows finding good priority functions with fewer functions processed (generated and evaluated) by avoiding prompts that include functionally identical examples differing only in syntax (see Appendix D).

### 4.5 Island initialization and resets

Each island is initialized with the same trivial priority function shown in Figure 2, which assigns equal priority to all sequences.

To allow information exchange between islands, we periodically reset them. During a reset, we discard the stored priority functions in the worst-performing half of islands, where performance is measured by the highest cluster score on each island. We then re-initialize each discarded island with the best-performing priority function from a randomly selected surviving island. We reset islands after every $R$ stored functions rather than after a fixed time interval as in Romera-Paredes et al. (2024), to separate the reset logic from the rate at which functions are processed.

Table 1: Code sizes for single-deletion correction. Each row corresponds to a run configuration: trivial initialization ($f^T$); first successful function after 120K processed ($f^{120K}$); best function from standard runs with varying hyperparameters ($f$); using weighted scoring ($f^W$); prompts 3 and 4 with StarCoder2 ($f^{3,4}$) and GPT-4o mini ($f^{3,4/\text{GPT}}$). Bold indicates the $\text{VT}_0(n)$ bound, which is optimal for $n \leq 11$.

| Priority function | $n = 6$ | 7 | 8 | 9 | 10 | 11 | 12 | 13 | 14 | 15 | 16 |
|---|---|---|---|---|---|---|---|---|---|---|---|
| $f^T$ | 8 | 14 | 25 | 42 | 71 | 125 | 224 | 406 | 737 | 1345 | 2468 |
| $f^*$ | **10** | **16** | **30** | **52** | **94** | **172** | **316** | **586** | 1054 | 2000 | 3389 |
| $f^{120K}$ | **10** | **16** | **30** | **52** | **94** | **172** | **316** | 449 | 794 | 1386 | 2515 |
| $f^{W*}$ | **10** | **16** | **30** | **52** | **94** | **172** | **316** | 564 | **1096** | 1364 | 2493 |
| $f^{3,4\&3/\text{GPT}}$ | **10** | **16** | **30** | **52** | **94** | **172** | **316** | **586** | **1096** | **2048** | **3856** |
| $f^{4/\text{GPT}}$ | **10** | **16** | **30** | **52** | **94** | **172** | **316** | **586** | 1083 | 2025 | 3696 |

*Reported code sizes are not constructed by a single priority function. For each code length $n$, we report the maximum size achieved across all successful functions discovered within the run configuration.

## 5 EXPERIMENTS

We run 20 evolutionary search experiments, varying the initial temperature $T$, sampling period $P$, and the number of functions $R$ stored before an island reset. Each experiment runs with or without dynamically decreasing the LLM sampling temperature to balance exploration and exploitation.

Our main finding is that FunSearch discovers priority functions that construct maximum-size single-deletion-correcting codes for lengths $6 \leq n \leq 11$, including previously unknown constructions. For longer code lengths ($n > 11$), where VT codes are conjectured to be optimal, FunSearch rediscovers them within our greedy framework and also finds alternative constructions of the same size (verified up to length $n = 25$). For two deletions, we discover larger codes than previously known for code lengths $n = 12, 13$ and $16$.

### 5.1 EXPERIMENTAL SETUP

We evaluate the generated priority functions on code sizes achieved for a single deletion ($s = 1$) and lengths $n \in [6, 11]$, where the maximum independent set sizes are known. The evaluation range balances computational feasibility and problem difficulty. Smaller code lengths $n$ make the problem trivial, while larger $n$ result in prohibitive computational and memory costs.

Each evolutionary search processes up to 400K priority functions, which takes about 350 GPU hours. Performance is measured as a binary outcome: success or failure. A function is considered successful if it constructs maximum independent sets on all evaluation inputs. A configuration is successful if it discovers at least one successful function. If a run succeeds before 400K functions, we stop the search early. We then generate and evaluate an additional 20K functions to find others that may generalize better to longer code lengths.

Testing priority functions on large code lengths is computationally expensive because the number of sequences grows exponentially with code length. We therefore evaluate functions on shorter code lengths where optimal sizes are known during the search. At the end of each search, we test only those achieving optimal sizes on longer code lengths to identify priority functions that generalize.

We use the LLM hyperparameters listed in Table 7c in Appendix B, which we find perform best in smaller-scale experiments.

### 5.2 UNDERLYING LOGIC OF PRIORITY FUNCTIONS

We first identify common logical structures in the discovered priority functions and then discuss their relation to the best known VT codes. We categorize the discovered priority functions into graph-based and number-theoretic functions.

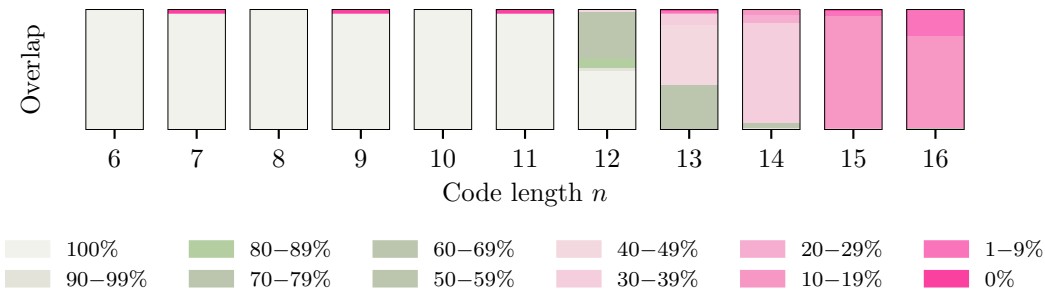

Figure 3: Sequence overlap between discovered priority functions and the largest $VT_0(n)$ codes for $n \in [6, 16]$. Color denotes overlap bin and bar height the number of functions.

Graph-based priority functions assign priority based on local graph connectivity and sequence characteristics, considering both the degree of a vertex and the bit patterns of its neighbors. An example is in Figure 10 in Appendix C.

Number-theoretic priority functions assign priority based on the integer representations of neighboring sequences and their bit patterns. An example is in Figure 11 in Appendix C.

The best-known single-deletion-correcting codes are VT codes (Varshamov & Tenengolts, 1965). For code length $n$ and parameter $a \in \mathbb{Z}$, we denote by $VT_a(n)$ the set of binary sequences $\mathbf{v} \in \{0, 1\}^n$ satisfying

$$\sum_{i=1}^{n} i v_i \equiv a \pmod{n + 1}, \tag{1}$$

where each bit $v_i$ is weighted by its position $i$ in the sequence.

The $VT_0(n)$ code has maximum code size as $n \rightarrow \infty$ and is conjectured to have maximum code size for all code lengths $n$. In our framework, $VT_0(n)$ codes can be represented by a priority function that assigns a high priority (e.g., $+\infty$) to sequences satisfying Equation 1 with $a = 0$, and a low priority (e.g., 0) to those that do not.

Figure 3 shows the sequence overlap between the codes constructed by our discovered priority functions and the largest $VT_0(n)$ codes for code lengths $n \in [6, 16]$. Many of our discovered priority functions recover the largest $VT_0(n)$ codes with 100% sequence overlap and follow similar logic, i.e., they assign weights to bits based on their position in the sequence. However, priority functions that use graph structure alongside sequence information discover previously unknown codes. For example, the graph-based priority function in Figure 10 (Appendix C) achieves the same code size as the largest $VT_0(n)$ codes (verified up to $n = 25$), but shares no sequences with $VT_0(n)$ for $n = 7, 9, 11$, and 13.

### 5.3 GENERALIZATION TO LONGER CODE LENGTHS AND MULTIPLE DELETIONS

Our approach searches for priority functions that construct deletion-correcting codes, rather than searching for the codes directly. This allows us to construct longer and multiple deletion-correcting codes with the priority functions found for short code lengths and a single deletion.

Table 1 shows that priority functions optimized for code lengths $n \in [6, 11]$ also achieve the conjectured largest $VT_0(n)$ code sizes for $n = 12, 13$ and remain close for $n \in [14, 16]$.

For two deletions, the priority functions construct codes whose sizes are close to the best known over the tested lengths $n \in [7, 16]$. For $n = 13$, our search discovers a two-deletion-correcting code of size 50, improving on the previous best known size of 49. The corresponding priority function is shown in Figure 9, and detailed results are given in Table 4 (Appendix I).

Our approach outperforms previous search-based methods (Landjev & Haralambiev, 2007; Swart & Ferreira, 2003) that search the full space of $2^n$ binary sequences on code lengths $n \in [12, 16]$ for two deletions. Searching the sequence space becomes exponentially harder with the code length,

```
def f(v, G, n, s):
    # The condition ord(a) > 125 has no effect, as the ASCII values of '0' and '1' are always below 125.
    v = ''.join(['-' * (ord(a) > 125) + a for a in list(v)])
    onepositions = [c for c, d in reversed(list(enumerate(v, start=-len(v)))) if d == '1']
    negonesum = sum([-c for c in onepositions])
    # Maximum of negonesum is (n-1)/2 for n odd and n/2 for n even, which is always < n, so taking mod n does
      not change the priority
    finalans = (⌊negonesum/((n + s) · 1)⌋ % n)
    return finalans
```

Figure 4: Priority function generated using prompt 4 (Figure 20) that constructs the $VT_0(n)$ code for $s = 1$. Comments added for clarity.

making it increasingly difficult to discover large codes. In contrast, our approach searches in the space of priority functions, independent of code length.

These results show that priority functions optimized for single-deletion correction can generalize to longer code lengths and, to some extent, to multiple deletions. However, we did not find a priority function that achieves maximum single-deletion code sizes (where known) and matches or exceeds best-known sizes for two deletions.

### 5.4 PROMPT ENGINEERING AND GENERAL-PURPOSE LLMS

To assess whether prompt engineering improves generalization to longer code lengths or sample efficiency (fewer functions processed before success), we modify the baseline prompt in Figure 2. We also test GPT-4o mini, an instruction-tuned model trained on diverse tasks beyond code generation, which may better interpret the task than code-only models.

We find that prompt engineering improves generalization for both StarCoder2 and GPT-4o mini and improves sample efficiency for GPT-4o mini. Explicitly instructing StarCoder2 to consider binary string properties leads to rediscovering the largest $VT_0(n)$ codes in an alternative form.

#### 5.4.1 PROMPT ENGINEERING

We test five prompts. Prompt 1 explicitly states that we are considering the single deletion case ($s = 1$) and that the priority function determines the importance of each vertex for inclusion in the independent set. Prompt 2 includes the evaluation script to provide context on how the priority function determines independent set size through greedy selection. Prompt 3 removes the graph $G$ as an input to the priority function and excludes the `networkx` package to bias the LLM toward computing priority based on sequence structure only. Prompt 4 explicitly instructs the LLM to consider sequence structure. Prompt 5 combines modifications from prompts 1 and 4. The prompts are shown in Appendix G.1.

Table 1 shows that the priority functions discovered using StarCoder2 with prompts 3 and 4 generalize better to longer code lengths. Figures 19 and 21 in Appendix G.1 show examples of priority functions found with prompts 3 and 4, respectively, that achieve $VT_0(n)$ code sizes for all tested code lengths $n \in [6, 25]$, but follow a different logic. The function in Figure 21 constructs new codes for odd lengths that have zero sequence overlap with the largest $VT_0(n)$ codes in this range. Figure 4 shows the priority function found with prompt 4, which is equivalent to the largest $VT_0(n)$ codes for all code lengths, as explained in Appendix H.

The other prompts fail to find successful priority functions within 400K processed. With prompt engineering (prompt 3), the first successful function is discovered after approximately 300K functions, compared to 120K in the best run without prompt engineering. This suggests that, for StarCoder2, the prompts considered here do not improve sample efficiency.

#### 5.4.2 GPT-4O MINI FOR GENERATING PRIORITY FUNCTIONS

Figure 5 shows that GPT-4o mini finds a successful priority function with fewer candidates than StarCoder2 (69K vs. 120K) and generates a larger fraction of executable functions (43.7% vs. 16.2%). However, without prompt engineering, GPT-4o mini fails to find successful functions within 400K processed. Successful solutions are only found with prompts 3 and 4.

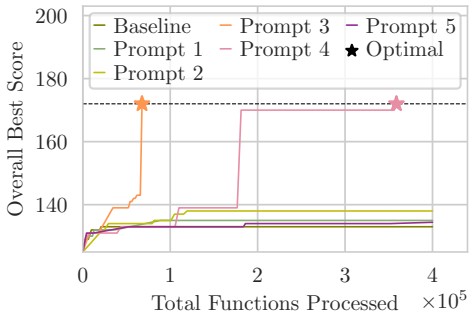
(a) Search trajectory with GPT-4o mini.

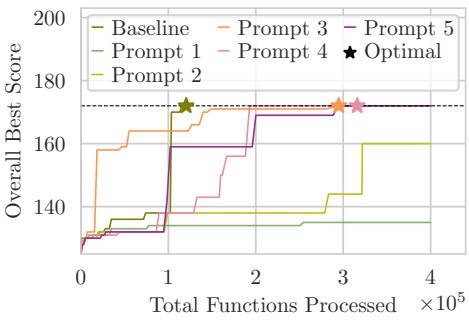
(b) Search trajectory with StarCoder2.

Figure 5: GPT-4o mini finds successful priority functions with fewer processed and generates more executable functions than StarCoder2, but requires prompt engineering.

Figures 23 and 24 in Appendix G.2 show examples of priority functions discovered with GPT-4o mini using prompt 3 and prompt 4, respectively. Functions generated with prompt 3 achieve 100% sequence overlap with the largest $VT_0(n)$ codes for lengths $n \in [6, 25]$, while functions generated with prompt 4 achieve $VT_0(n)$ code sizes for $n \in [6, 13]$ and are close to $VT_0(n)$ code sizes for larger lengths $n \in [14, 16]$.

### 5.5 SEARCH FOR MULTIPLE DELETION-CORRECTING CODES

We now conduct evolutionary searches for two-deletion-correcting codes. Since optimal code sizes are unknown in this regime, we process all 400K functions without early stopping and analyze all functions in the program database that achieve a larger average size on the evaluation inputs than the trivial initialization.

We consider two additional evaluation sets for the search. The first scores functions on two-deletion-correcting code sizes for $n \in [7, 12]$. The second jointly scores single- and two-deletion correction, using $n \in [9, 11]$ for $s = 1$ and $n \in [10, 12]$ for $s = 2$. Each set runs with the default configuration from Section 5.1, as well as weighted scoring and prompt 4, totaling six additional runs.

Searches targeting two-deletion correction discover a new lower bound at $n = 12$, improving from 32 to 34 (e.g., Figure 28). The joint search finds a new bound at $n = 16$, improving from 201 to 204 and functions achieving $VT_0(n)$ sizes for single deletion with $n \in [6, 13]$ that closely match best-known sizes for two and three deletions over $n \in [7, 16]$ (e.g., Figure 33). Appendix I provides details, Table 4 summarizes achieved sizes, and Figure 26 shows differences from best-known sizes.

## 6 CONCLUSION AND LIMITATIONS

In this work, we used LLMs and evolutionary search to discover new deletion-correcting codes and re-discover existing ones. While we focused on deletions, our method can be applied to any error type or combination thereof. We showed that priority functions optimized for short code lengths and a single deletion can generalize to longer lengths and, to some extent, to multiple deletions.

A key limitation of our approach is the poor scalability of the evaluator, which makes evolutionary search infeasible for moderate to large code lengths. The evaluator must compute priorities for exponentially many sequences as code length increases. For graph-based priority functions, the evaluator must also construct or load the full graph that stores all sequences and pairwise edges, which quickly becomes memory-prohibitive.

Nonetheless, searching in function space generalizes better than previous approaches (Landjev & Haralambiev, 2007; Swart & Ferreira, 2003) that search over all sequences, and our priority functions can be mathematically analyzed to determine code sizes without explicit construction, as we demonstrated with the priority function that rediscovered VT codes.

## LLM USAGE

Large language models were used as writing assistance tools for editing and polishing the text for this submission.

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

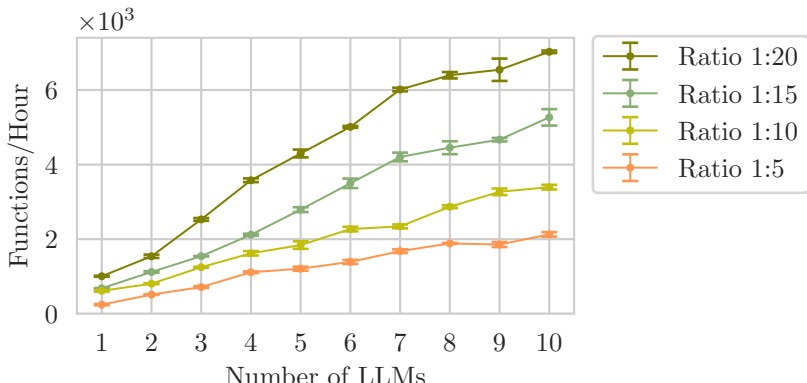

Figure 6: Rate at which functions are processed for different LLM-to-evaluator ratios in our distributed implementation of FunSearch.

## A  IMPLEMENTATION DETAILS

We implement FunSearch using RabbitMQ (Pivotal Software) for parallelization via asynchronous message passing. The system consists of multiple LLMs and evaluators, and a single program database, each running as an independent worker. Workers communicate through RabbitMQ queues using the Advanced Message Queuing Protocol (AMQP) 0-9-1, which runs over the Transmission Control Protocol (TCP). Each worker consumes and publishes messages to their designated queues.

The program database constructs prompts and sends them to the LLM queue. The LLMs process these prompts to generate new priority functions, which are published to the evaluator queue. The evaluators compute evaluation scores and return the results to the program database queue.

The number of functions that can be processed within a fixed time interval is determined by the number of LLMs and evaluators. We run our implementation of FunSearch with different LLM-to-evaluator ratios to understand how resource allocation affects throughput. Each LLM runs on a single GPU (NVIDIA A100 (80GB) or H100 (94GB)), while each evaluator processes inputs in parallel using two CPU cores. Evaluators execute functions with a 5-minute timeout; if execution exceeds this limit, the function is considered non-executable.

Figure 6 shows the throughput in functions per hour (higher is better) for different LLM-to-evaluator ratios. We achieve the highest throughput at the largest tested ratio of 20 evaluators per LLM. We expect that increasing the number of evaluators further would increase throughput, but we could not test this due to infrastructure constraints. The reported results correspond to a suboptimal setup where evaluators construct the graph from scratch rather than loading a precomputed file, which increases evaluation time. Using precomputed graphs increases throughput further, but does not change the conclusion that evaluators are the limiting factor, and increasing their number relative to LLMs increases throughput up to a point.

If processing rates between LLMs and evaluators are imbalanced during execution, our implementation also supports dynamically scaling their number (within available resources) to optimize throughput.

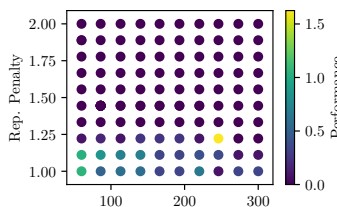 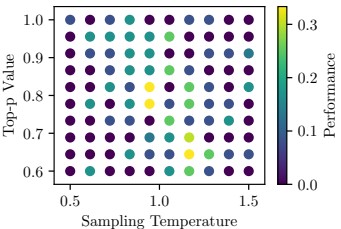

| Parameter | Best | Range |
|---|---|---|
| Rep. Penalty | 1.2 | [1,2] |
| Top-p | 0.78 | [0.6,1] |
| Max. Tokens | 246 | [50,300] |
| Temp. | 0.94 | [0.5,1.5] |

(a) Performance across maximum new tokens and repetition penalty.

(b) Performance across temperature and top-p.

(c) Best-performing hyperparameters.

Figure 7: Results of LLM hyperparameter optimization from smaller-scale experiments.

## B  LLM HYPERPARAMETER OPTIMIZATION

We conduct two independent grid searches for the LLM hyperparameters, varying maximum new tokens and repetition penalty while keeping temperature and top-p fixed, and vice versa.

We measure performance as the average improvement in the independent set sizes constructed by the best priority functions across all islands for all code lengths $n \in [6, 11]$ with deletion parameter $s = 1$, relative to the trivial initialization. Each grid search run is evaluated after one hour using one GPU and 40 CPUs to balance search depth with computational feasibility.

For the grid search over maximum new tokens, we consider values in the range $[60, 300]$, and for repetition penalty, values in $[1.0, 2.0]$, both divided into 10 equally spaced grid points. Temperature and top-p are fixed at 0.2 and 0.95, respectively, as in Section 7.1.3 of Lozhkov et al. (2024). The results are shown in Figure 7a. Low repetition penalties combined with high maximum new tokens often result in the LLM repeating the code completion task, generating multiple function headers with minor variations or trivial return statements instead of a single, improved function. Repetition penalties above 1.22 fail to generate executable functions. While competitive results are achieved with maximum new tokens between 60 and 140 and repetition penalties between 1.05 and 1.11, the highest performance is observed with 246 maximum new tokens and a repetition penalty of 1.22. As discovering new maximum code sizes requires only a single priority function, we proceed with these hyperparameters.

For the grid search over temperature and top-p, we consider values in $[0.5, 1.5]$ and $[0.6, 1.0]$, respectively, with 10 equally spaced grid points, while keeping maximum new tokens fixed at 246 and the repetition penalty at 1.22. The results are shown in Figure 7b. Higher variability in token sampling (larger temperature and top-p values) increases fluctuations in the performance metric but also improves performance. More deterministic sampling results in more syntactically correct functions but does not lead to better performance.

These findings align with the hypothesis of Romera-Paredes et al. (2024) that the LLM contributes by exploring diverse function solutions, occasionally generating good executable functions but often producing unusable outputs. The best performance is achieved at a temperature of 0.9444 and a top-p of 0.7778.

Table 2: Results for different evolutionary search hyperparameter configurations. A check mark ($\checkmark$) indicates that the configuration discovered a priority function achieving the maximum code size; a cross ($\times$) indicates it did not.

(a) Results for initial temperature $T$, with $P = 30K$ and $R = 1.2K$ fixed.

| $T$ | $n = 6$ | $n = 7$ | $n = 8$ | $n = 9$ | $n = 10$ | $n = 11$ |
|------|------|------|------|------|------|------|
| 0.05 | $\times$ | $\checkmark$ | $\times$ | $\times$ | $\times$ | $\times$ |
| 0.1 | $\checkmark$ | $\checkmark$ | $\checkmark$ | $\checkmark$ | $\checkmark$ | $\checkmark$ |
| 0.3 | $\checkmark$ | $\checkmark$ | $\times$ | $\times$ | $\checkmark$ | $\times$ |
| 0.5 | $\times$ | $\times$ | $\checkmark$ | $\times$ | $\times$ | $\times$ |
| 1 | $\checkmark$ | $\checkmark$ | $\times$ | $\times$ | $\times$ | $\times$ |

(b) Results for period $P$, with $T = 0.1$ and $R = 1.2K$ fixed.

| $P$ | $n = 6$ | $n = 7$ | $n = 8$ | $n = 9$ | $n = 10$ | $n = 11$ |
|------|------|------|------|------|------|------|
| 5,000 | $\checkmark$ | $\checkmark$ | $\checkmark$ | $\checkmark$ | $\checkmark$ | $\checkmark$ |
| 10,000 | $\checkmark$ | $\checkmark$ | $\checkmark$ | $\times$ | $\times$ | $\times$ |
| 30,000 | $\checkmark$ | $\checkmark$ | $\checkmark$ | $\checkmark$ | $\checkmark$ | $\checkmark$ |
| 50,000 | $\checkmark$ | $\checkmark$ | $\checkmark$ | $\checkmark$ | $\checkmark$ | $\checkmark$ |
| 100,000 | $\checkmark$ | $\checkmark$ | $\checkmark$ | $\times$ | $\times$ | $\times$ |

(c) Results for the number of functions $R$ stored before an island reset, with $T = 0.1$ and $P = 30K$ fixed.

| $R$ | $n = 6$ | $n = 7$ | $n = 8$ | $n = 9$ | $n = 10$ | $n = 11$ |
|------|------|------|------|------|------|------|
| 300 | $\checkmark$ | $\checkmark$ | $\checkmark$ | $\checkmark$ | $\checkmark$ | $\checkmark$ |
| 600 | $\checkmark$ | $\checkmark$ | $\checkmark$ | $\checkmark$ | $\checkmark$ | $\checkmark$ |
| 1200 | $\checkmark$ | $\checkmark$ | $\checkmark$ | $\checkmark$ | $\checkmark$ | $\checkmark$ |
| 2400 | $\times$ | $\checkmark$ | $\times$ | $\times$ | $\times$ | $\times$ |
| 5000 | $\checkmark$ | $\checkmark$ | $\times$ | $\checkmark$ | $\times$ | $\times$ |

(d) Results for dynamically decreasing the LLM temperature to greedy decoding after storing $D$ functions.

| $D$ | $n = 6$ | $n = 7$ | $n = 8$ | $n = 9$ | $n = 10$ | $n = 11$ |
|------|------|------|------|------|------|------|
| 5,000 | $\checkmark$ | $\checkmark$ | $\checkmark$ | $\checkmark$ | $\checkmark$ | $\checkmark$ |
| 10,000 | $\times$ | $\checkmark$ | $\times$ | $\checkmark$ | $\times$ | $\times$ |
| 20,000 | $\checkmark$ | $\checkmark$ | $\checkmark$ | $\checkmark$ | $\checkmark$ | $\times$ |
| 50,000 | $\checkmark$ | $\checkmark$ | $\times$ | $\times$ | $\checkmark$ | $\checkmark$ |

## C  EVOLUTIONARY SEARCH HYPERPARAMETER OPTIMIZATION

We perform independent grid searches over the evolutionary search hyperparameters initial temperature $T$, sampling period $P$ and the number of functions $R$ stored before an island reset, using the best-performing LLM hyperparameters from Table 7c. Performance is measured as a binary outcome: success or failure in finding a priority function that constructs a maximum independent set for all evaluation inputs $n \in [6, 11]$ with $s = 1$, where the maximum is known. Each evolutionary search run is evaluated after generating 400K priority functions or stops early if a successful function is found and 20K additional ones are generated. Examples for graph-based and number-theoretic functions are given in Figures 10 and 11, respectively.

Table 2a summarizes the results for initial temperatures $T \in \{0.05, 0.1, 0.3, 0.5, 1\}$ with a fixed sampling period of $P = 30K$ and $R = 1.2K$ functions stored before a reset. A successful priority function is found only when the temperature is set to $T = 0.1$. Figure 8a shows the evolutionary search trajectories, plotting the highest score assigned to priority functions across all clusters and islands as new functions are processed. With $T = 0.1$, a successful function (shown in Figure 9) is found after approximately 115,850 processed functions, with 20.7% of generated functions stored at the end of the search. When the temperature is set to $T = 0.05, 0.3, 0.5,$ or $1$, the percentages of stored functions are 18.6%, 19.3%, 12.0%, and 10.0%, respectively. Across all configurations, only a small fraction of the generated functions are stored, with many failed executions.

Table 2b summarizes the results for sampling periods $P \in \{5K, 10K, 30K, 50K, 100K\}$, with a fixed temperature of $T = 0.1$ and $R = 1.2K$ functions stored before a reset. Adjusting the sampling period does not improve performance beyond the configuration with $P = 30K$ in the grid search over temperature. Figure 8b shows the evolutionary trajectories for different sampling periods. With $P = 5K$, a successful priority function is found after 193,815 processed functions, with 18.1% stored at termination. With $P = 50K$, a successful function is found after 132,499 processed functions, with 23.0% stored. When the sampling period is set to $P = 10K$ or $P = 100K$, no successful function is found after 400K processed functions, and the fractions of stored functions are 13.0% and 19.8%, respectively.

Table 2c summarizes the results for numbers of functions $R \in \{300, 600, 1.2K, 2.4K, 5K\}$ stored before an island reset, with a fixed temperature of $T = 0.1$ and a sampling period of $P = 30K$. Varying $R$ does not improve performance beyond the configuration with $R = 1.2K$ in the grid search over temperature. Figure 8c shows the evolutionary trajectories for different values of $R$. With $R = 300$, a successful priority function is found after 251,359 processed functions, with 18.2% stored at termination. With $R = 600$, a successful function is found after 196,756 processed

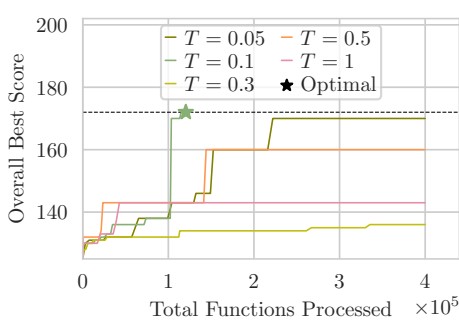

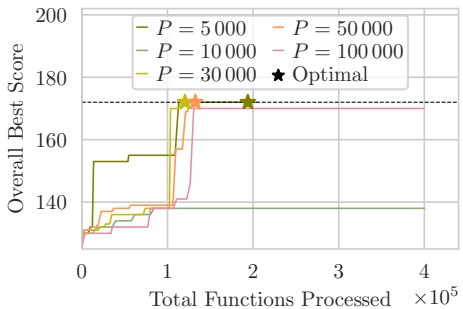

(a) Varying initial temperature $T \in \{0.05, 0.1, 0.3, 0.5, 1\}$ with fixed $P = 30K$ and $R = 1.2K$.

(b) Varying $P \in \{5K, 10K, 30K, 50K, 100K\}$ with fixed $T = 0.1$ and $R = 1.2K$.

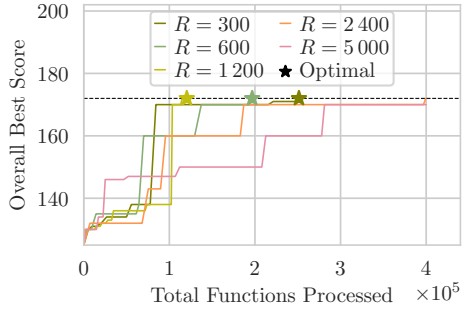

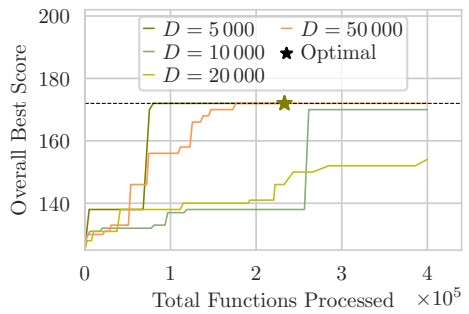

(c) Varying number of functions $R \in \{300, 600, 1.2K, 2.4K, 5K\}$ stored before an island reset, with fixed $T = 0.1$ and $P = 30K$.

(d) Dynamically decreasing LLM temperature, reaching greedy decoding at $D \in \{5K, 10K, 20K, 50K\}$ functions.

Figure 8: Trajectories for varying evolutionary search hyperparameters.

functions, with 19.9% stored. When $R = 2,400$ or $R = 5K$, no successful function is found within 400K processed, and the fractions of stored functions are 19.2% and 19.6% , respectively.

We also experiment with dynamically decreasing the LLM sampling temperature to balance exploration and exploitation. The temperature is initialized at $0.94$ and decreases as more functions are stored on the island from which the prompt is sampled, reaching zero at $D \in \{5K, 10K, 20K, 50K\}$ stored functions. Similar to reducing the temperature for sampling clusters as more functions are stored, decreasing the LLM sampling temperature makes token sampling more deterministic over time, promoting the exploitation of higher-scoring function examples in prompts.

Table 2d summarizes the results for dynamically decreasing the LLM sampling temperature for different values of $D$. While this approach slightly increases the number of executable functions, it does not improve search efficiency in finding a successful priority function with fewer functions processed compared to a fixed temperature. Figure 8d shows the evolutionary trajectories. With $D = 5K$, a successful priority function is found after 246,639 processed functions, with 22.6% stored at termination. When $D = 10K, 20K$, or $50K$, no successful function is found within 400K processed, with 21.1%, 17.2%, and 21.4% stored, respectively.

Table 3: Evolutionary search configurations that find successful priority functions with 400K processed.

| Initial $T$ | Period $P$ | Reset $R$ | Dynamic $D$ |
|---|---|---|---|
| 0.1 | 30,000 | 1,200 | w/o |
| 0.1 | 30,000 | 1,200 | 5,000 |
| 0.1 | 30,000 | 300 | w/o |
| 0.1 | 30,000 | 600 | w/o |
| 0.1 | 5,000 | 1,200 | w/o |
| 0.1 | 50,000 | 1,200 | w/o |

```
def f(v, G, n, s):
    neighbours = []
    for neighbor in G[v]:
        p = np.log(int(neighbor[:-s], 2) + 1) * \
            (2 ** (((len(neighbor) - s) - neighbor[:(-s)].count('0')) +
                ((neighbor[-s:] != '0') * len([i for i in range(0, len(neighbor), 8)])))) / \
            np.exp(sum([(i == "1") * len([j for j in ["1"] * 3]) for i in neighbor]))
        neighbours.append((p, neighbor))
    if not neighbours:
        return 0
    return sorted(neighbours, key=lambda x: x[0], reverse=True)[0][0]
```

Figure 9: Successful priority function $f^{120K}$ found after about 120K processed with $T = 0.1$, $P = 30K$ and $R = 1.2K$.

```
def f(v, G, n, s):
    position = [(j + 1) · (n - j)/(6 · s) for j, value in enumerate v if int(value) == 1]
    total_position = np.sum(position)
    degree = G.degree(v)/ float(n)
    return 4 · total_position + 5 · degree
```

Figure 10: Graph-based priority function that constructs codes with zero sequence overlap with the largest $\text{VT}_0(n)$ codes for lengths $n = 7, 9, 11, 13$ while achieving the same code size.

```
def f(v, G, n, s):
    def _find_matches(vertex, n, s):
        counter = 0
        counter = sum ([int(c) · (2^i - 1) for i, c in enumerate(reversed(list(vertex)))])
        return (bin(counter)).count("1")
    def _count_ones(vertex):
        counter=0
        counter=sum([int(_) for _ in list(vertex)])
        return counter
    weights=[(_find_matches(vertex_, n, s)/(s+0.5)*np.exp(-(_count_ones(vertex_))), vertex_) for vertex_ in G[
      v]]
    return sorted(weights)[-1]
```

Figure 11: Number-theoretic priority function that constructs the same codes as the largest $\text{VT}_0(n)$ codes for lengths $n \in [6, 11]$, but follows a different logic.

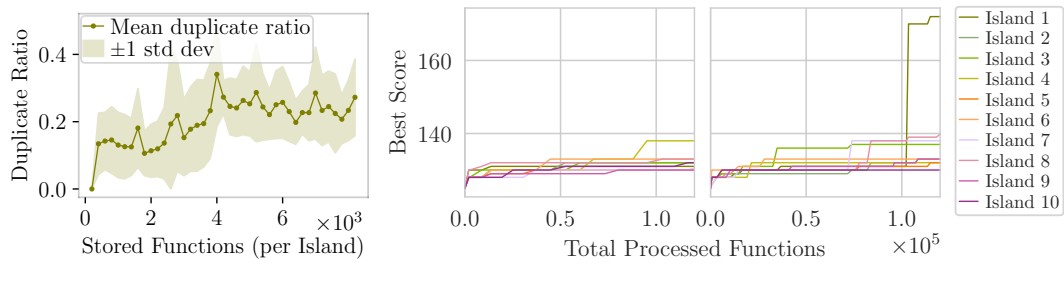

(a) Mean duplicate ratio.    (b) Best score trajectory without (left) and with (right) deduplication.

Figure 12: Results of the evolutionary search experiments with and without function deduplication.

# D    EFFECTIVENESS OF DEDUPLICATION

We assess the effectiveness of our function deduplication step by running an evolutionary search without it. For this search, we use an initial temperature $T = 0.1$, a sampling period $P = 30K$, store $R = 1.2K$ functions before an island reset, and use the best-performing LLM hyperparameters listed in Table 7c.

Figure 12a shows the mean duplicate ratio, defined as the number of duplicate functions divided by the total number of stored functions. It is computed every 200 stored functions per island and averaged across all islands. Without deduplication, approximately 20% of stored functions are duplicates, either exact copies of previously seen priority functions or syntactically modified versions with unchanged logic.

Figure 13 shows a prompt with duplicate few-shot examples and the corresponding function output generated by the LLM. Duplicate examples often result in the LLM generating functions with identical logic or minor syntactic changes, which fail to construct larger independent sets.

Figure 12b compares the run without deduplication to the run with deduplication that uses the same run configurations. The search with deduplication processes fewer functions before finding a successful one, suggesting that deduplication improves sample efficiency by filtering out prompts with duplicate examples.

```python
import numpy as np
import networkx as nx

def f_0(v, G, n, s):
    """Returns the priority with which we want to add vertex v."""
    vertex = str(v)
    count = 0
    for j in range(3, n + 1):
        for k in range(n - (j - 1)):
            ones = []
            zeros = []
            if vertex[k] == '1' or vertex[k] == '0':
                for l in range(k, k + j):
                    if vertex[l] == '1':
                        ones.append("1")
                    elif vertex[l] == '0':
                        zeros.append("0")
                if len(ones) > len(zeros):
                    count += len(ones)
                    ones = []
                elif len(zeros) > len(ones):
                    count -= len(zeros)
                    zeros = []
                else:
                    pass
    return count

def f_1(v, G, n, s):
    """  Improved version of f_0."""
    vertex = str(v)
    count = 0
    for i in range(3, n + 2):
        # print("i is ")
        # print(i)
        if i <= n + 1:
            for j in range(n - (i - 1)):
                ones = []
                zeros = []
                for k in range(j, j + i):
                    if vertex[k] == '1':
                        ones.append("1")
                    elif vertex[k] == '0':
                        zeros.append('0')
                if len(ones) > len(zeros):
                    count += len(ones)
                elif len(ones) < len(zeros):
                    count -= len(zeros)
    return count

def f_2(v, G, n, s):
    """Improved version of f_1."""
    vertex = str(v);
    count = 0;
    ones = list();
    zeros = list ();
    for i in range((n)):
        if vertex[i] == '0':
            zeros.append("0");
        if vertex[i] == '1':
            ones.append ("1")
        if (len(ones)-len(zeros)) >= 1 :
            count+= len(ones);
            while len(ones)!=0:
                ones.pop(-1)
            zeros=[]
        elif (-len(ones)+len(zeros))>= 1:
            count-= len(zeros);
            while len(zeros)!=0:
                zeros.pop(-1)
            # ones=[];
    return int(count /4 )
```

Figure 13: Prompt with duplicate few-shot examples $f_0$ and $f_1$ and the function $f_2$ generated by the LLM.

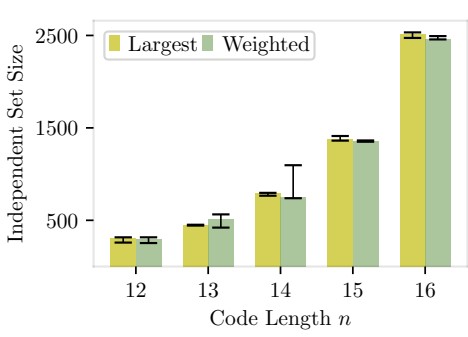 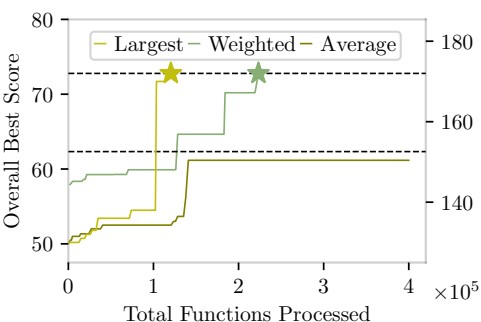

(a) Average independent set size for code lengths beyond the evaluation range, computed over all priority functions, with error bars showing the minimum–maximum range.

(b) Search trajectories. The dotted lines indicate the maximum scores at 172 (right axis), 72.78 (left axis), and 62.33 (left axis) for largest, weighted, and average scoring, respectively.

Figure 14: Results of evolutionary searches with different scoring functions.

```
def f(v, G, n, s):
    return -np.average([float(((int(y[:n-(s+1)].count('1'))*( int((y[-1:( -(n-s)):(-1)]).count ('1') )))**2/
    len(list(G.neighbors(y))))) for y in [ v ]+(list(G.neighbors(v)))])
```

Figure 15: Priority function $f^W$ found using weighted scoring.

## E    EFFECT OF THE SCORING FUNCTION ON PERFORMANCE AND GENERALIZATION

The experiments in Section 5.2 of the main paper show that the priority functions discovered using the baseline prompt generalize to code lengths $n = 12, 13$, beyond the evaluation range $n \in [6, 11]$, but remain only close to the largest $\mathrm{VT}_0(n)$ code sizes for larger code lengths $n$.

To improve generalization to longer code lengths, we explore aggregate scoring functions that evaluate priority functions based on their performance across all code lengths in the evaluation range, rather than only on the largest length. We compare two aggregate scoring strategies against the baseline, which uses the independent set size at length $n = 11$. The first is a simple average of independent set sizes over all evaluated lengths ($n \in [6, 11]$). The second is a weighted average over the same range, with weights proportional to $n$. All runs use an initial temperature $T = 0.1$, sampling period $P = 30K$, number of functions $R = 1.2K$ stored before an island reset, and the best-performing LLM hyperparameters listed in Table 7c.

Perhaps surprisingly, Figure 14a shows that the baseline scoring function achieves better generalization than the two aggregate alternatives. While the weighted scoring function discovers a priority function that achieves the largest $\mathrm{VT}_0(n)$ code size at $n = 14$, the baseline consistently finds functions that construct larger code sizes for all other tested lengths ($n \in [12, 16] \setminus \{14\}$). Figure 14b further shows that evaluating only on the largest code length finds a successful priority function with fewer processed than the weighted scoring function. In contrast, the average scoring function fails to find a successful function within 400K processed. These results suggest that focusing on the largest evaluated length is both more efficient and more effective for discovering functions that generalize to longer code lengths when searching for large single-deletion-correcting codes.

Given these findings, we also run an evolutionary search using only the largest code size $n = 11$ (and $s = 1$) to reduce computational overhead. However, evaluating priority functions on a single code length biases the search toward functions that are hardcoded for $n = 11$ and fail to execute for other lengths. Additionally, this setup affects clustering. Functions are now clustered based on their score (their performance on the largest code length $n = 11$) rather than their independent set sizes across all evaluated code lengths ($n \in [6, 11]$). This results in fewer, larger clusters (and thus fewer distinct function length ranges). As a result, shorter functions are sampled more frequently, and the few-shot prompts become less diverse compared to clustering based on multiple evaluation inputs.

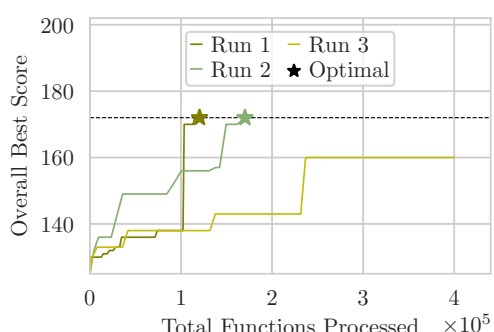

Figure 16: Trajectories for multiple runs with the same configuration using an initial temperature $T = 0.1$, sampling period $P = 30K$, and number of functions $R = 1.2K$ stored before an island reset. Two out of the three runs find a successful priority function within 400K processed.

## F    VARIATION ACROSS EVOLUTIONARY RUNS

The performance of FunSearch depends on two main factors: the quality of the LLM output and the functions sampled as examples for the few-shot prompt. These factors introduce inherent randomness into the method. To evaluate how FunSearch's performance varies across runs, we conduct two additional evolutionary search experiments with initial temperature $T = 0.1$, sampling period $P = 30K$, and $R = 1.2K$ functions stored before an island reset as well as the best performing LLM hyperparameters listed in Table 7c. This configuration previously found a successful function with the fewest processed.

Figure 16 shows the evolutionary search trajectories, plotting the maximum score (independent set size for the largest code length $n = 11$) as new functions are processed. Out of the three runs with the same configuration, two find a maximum independent set for all code lengths $n \in [6, 11]$ within the limit of 400K processed.

```
"""
Finds large independent set in graph G where vertices are binary strings of length n.
Vertices in G are connected if they share a subsequence of length at least n − s, where s = 1.

The functions f assign a priority to each vertex, indicating its importance for inclusion in the independent
    set.

Improve f₁ over its previous versions below.
Keep the code short and comment for easy understanding.
"""
import numpy as np
import networkx as nx

def f₀(v, G):
    """Returns the priority with which we want to add vertex v."""
    return 0.0

def f₁(v, G):
    """Improved version of f₀"""
```

Figure 17: Prompt 1 specifies the single-deletion case and explains that the priority function reflects the importance of each vertex for inclusion in the independent set.

```
"""
Finds large independent set in graph G where vertices are binary strings of length n.
Vertices in G are connected if they share a subsequence of length at least n − s.

Improve f₁ over its previous versions below.
Keep the code short and comment for easy understanding.
"""
import numpy as np

def f₀(v, G):
    """Returns the priority with which we want to add vertex v."""
    return 0.0

def f₁(v, G):
    """Improved version of f₀"""
```

Figure 18: Prompt 3 omits the graph $G$ as input to the priority function and removes the `import networkx as nx` line to bias the LLM toward computing priority based only on sequence structure.

## G    DETAILS ON PROMPT ENGINEERING AND GENERAL-PURPOSE LLMS

In this section, we provide additional details on prompt engineering and replacing StarCoder2 with GPT-4o Mini. For all runs, we use the configuration with an initial temperature $T = 0.1$, sampling period $P = 30K$, and number of functions $R = 1.2K$ stored before an island reset, as well as the best performing LLM hyperparameters as listed in Table 7c.

### G.1    PROMPT ENGINEERING

Here we describe our modifications to the baseline prompt in Figure 2. For prompts 3 and 4, which discover priority functions that achieve maximum code sizes where known, we further analyze their logic, with prompt 4 rediscovering the $\mathrm{VT}_0(n)$ code.

**Prompt 1** in Figure 17 specifies that we consider the single-deletion case and that priority reflects a vertex's importance for inclusion in the independent set. The rest remains identical to the baseline prompt.

We introduce prompt 1 after observing that many generated functions include redundant conditions when $s = 1$, such as $s > n$, which is always false. While explicitly stating $s = 1$ reduces such redundancies, it does not improve performance in constructing maximum independent sets.

**Prompt 2** in Figure 25 includes the entire evaluation script to give context on how the priority function is used to construct the independent set. The rest remains identical to the baseline prompt. Within the 400K processed functions, prompt 2 does not find a successful one. This may be because the additional context distracts from the main task of improving the priority function to construct larger independent sets.

```
def f(v, n, s):
    lst=[]
    for p in range ((n-2)) :
        for q in range (((p+2)),(n))    :
            string=""
            for r in range (p,q+1) :
                string+=v[r]
            lst.append(string)
    clist=[*map(lambda w:(w).count('1'),lst)]
    averageofobservations=(np.mean(clist));
    deviationfromaverage=(np.var(clist)**.65);
    priortiyvalue= -(averageofobservations/3+.3)*(deviationfromaverage**.65*(.7))+ (.8)+(1/(len(v)*2.5 ));
    return round(priortiyvalue,10)
```

Figure 19: Priority function found using prompt 3 that achieves largest $VT_0(n)$ code sizes for all evaluated lengths $n \in [6, 25]$ with 100% sequence overlap.

```
"""
Finds large independent set in graph G where vertices are binary strings of length n.
Vertices in G are connected if they share a subsequence of length at least n − s.

Improve f_1 over its previous versions below.
Keep the code short and comment for easy understanding.

Consider properties of the binary string v, such as specific patterns, the number of ones/zeros.
"""
import numpy as np
import networkx as nx

def f_0(v, G):
    """Returns the priority with which we want to add vertex v."""
    return 0.0

def f_1(v, G):
    """Improved version of f_0"""
```

Figure 20: Prompt 4 explicitly instructs the LLM to consider properties of the binary string, such as the number of zeros and ones.

**Prompt 3** in Figure 18 removes the graph $G$ as input to the priority function and the `network` package from the import statements to bias the LLM to generate functions that rely only on sequence-specific information. The rest remains identical to the baseline prompt.

The priority functions discovered using evolutionary search with prompt 3 follow a common structure. Most functions assign priority based on statistics of the number of 1-bits in an increasing sliding window over the sequence, with either a fixed minimum length (e.g., 2) or one determined by the deletion correction parameter $s$. The functions differ in which statistics of the 1-bit count they use (e.g., mean, variance, maximum) and how they transform the statistic(s) (e.g., scaling factors or number of unique sliding windows). These variations affect how well the priority function generalizes to longer code lengths. The function achieving the largest $VT_0(n)$ code sizes for lengths $n \leq 25$ is given in Figure 19, with 100% sequence overlap.

**Prompt 4** in Figure 20 explicitly instructs the LLM to focus on bit patterns in the sequence when assigning priority. The rest remains identical to the baseline prompt. As a result, StarCoder2 rediscovers the largest $VT_0(n)$ codes for all $n$. Beyond the VT formulation (discussed in Appendix H), the other discovered priority functions can be grouped into two main categories.

The first consists of functions that compute statistical properties of the sequence: the count of 1-bits, the product of their positions, and the sum of cumulative sums of 0-bit positions. The priority score is determined by applying bitwise operations (XOR, AND, OR, shifts) and logical conditions on these statistics, as illustrated in Figure 21. Interestingly, both categories have 100% overlap with the largest $VT_0(n)$ codes when $n$ is even and 0% overlap when $n$ is odd.

The second consists of a single function that assigns priority based on:

$$-\sum_{i=1}^{n} x_i \cdot (n - i + 1) \mod (n + 1) - b \mod n,$$

where $b = 1.5$. We find that this function appears multiple times with different values of $b$ but achieves maximum code sizes on the evaluation inputs only when $b = 1.5$. This suggests that the

```
def f(v, G, n, s):
    count_ones = np.array([int(char) for char in v]).sum()
    product_positions = abs((np.arange(n) * np.array([int(char) for char in v])).prod())
    sum_cumsum_zeros = ((~np.array([int(char) for char in v]).astype(bool)).cumsum().sum()) % (n + 1)
    c = [count_ones, product_positions, sum_cumsum_zeros]
    priority_score = min([
        ((c[-1] ** 4) & c[-2]) + (((c[-1] * 9) < c[-2])),
        ~((((-c[-1]) << c[-2]) ^ ~c[-1]) & ~c[-2]),
        ((~(~c[-2] | ~(c[-1])))) ^ (~c[-1]) ^ ((-(~(c[-1] | c[-2]))) ^ (c[-1] > 1)),
        ~(~c[-1] & ~c[-2]),
        (c[-1] + 1) == c[-2]
    ])
    return priority_score
```

Figure 21: Example of a priority function found using prompt 4 that achieves the largest $VT_0(n)$ code sizes for all evaluated code lengths $n \in [6, 20]$, based on statistical properties of the sequence. It has 100% sequence overlap for even $n$ and zero overlap for odd $n$.

LLM explores both globally and locally within the function space, even without being explicitly instructed to do so.

**Prompt 5** in Figure 22 combines the modifications of prompts 1 and 4. However, it does not find a successful priority function within 400K processed, even though prompt 4 rediscovers $VT_0$ codes. The rest remains identical to the baseline prompt.

```
"""
Finds large independent set in graph G where vertices are binary strings of length n.
Vertices in G are connected if they share a subsequence of length at least n − s.

The functions f assign a priority to each vertex v indicating its importance for inclusion in the independent
    set.

Desired properties of the function f:
- **Efficiency**: The function should be computationally efficient.
- **Avoid Redundant Computations**: Do not perform unnecessary calculations or repeat work.
- **Clarity**: The code should be easy to understand, with appropriate comments.
- **Innovation**: Explore different strategies for calculating the priority. Consider specific characteristics
        of the binary strings, such as:
    - Patterns in the binary string.
    - The number of ones or zeros (Hamming weight).
    - Distribution of bits (e.g., runs of ones or zeros).

Improve f_1 over its previous versions below.
Keep the code short and comment for easy understanding.
"""
import numpy as np
import networkx as nx

def f_0(v, G):
    """Returns the priority with which I want to add vertex v."""
    return 0.0

def f_1(v, G):
    """Improved version of f_0"""
```

Figure 22: Prompt 5 provides more detailed instructions, emphasizing efficiency, clarity, and innovation. It explains that priority reflects a vertex's importance for inclusion in the independent set, and prompts the LLM to consider binary string properties such as the number and distribution of zeros and ones.

### G.2 PRIORITY FUNCTIONS DISCOVERED WITH GPT-4O MINI

Here, we discuss the logic used by the priority functions discovered with GPT-4o Mini.

**Using Prompt 3.** The priority functions discovered with prompt 3 and GPT-4o mini follow a similar logic. They compute priority based on the counts of 1- and 0-bits, the number of 0-bits appearing after the last 1-bit, and the sum of 1-bits within certain sliding windows. Each function combines or weights these counts differently. An example is shown in Figure 23. These functions achieve the largest $VT_0(n)$ code sizes for all evaluated code lengths $n \leq 25$, with 100% sequence overlap.

**Using Prompt 4.** The priority functions discovered with prompt 4 and GPT-4o mini compute priority based on the number of 1- and 0-bits in a sequence, the count of 1-bits within sliding windows, and the number of neighbors each sequence has in the graph $G$. They differ primarily in how the

```
def f(v, n, s):
    ones_count = v.count('1')
    zero_count = v[:n - s].count('0')
    efficient_zero_contributions = sum(1 for i in range(n) if v[i] == '0' and '1' in v[:i])
    overlap_ones = sum(v[i:i + n - s].count('1') for i in range(n - s + 1))
    overlap_count = (overlap_ones + zero_count) // (n - s + 1)
    return ones_count + zero_count * (n - s + 2) + efficient_zero_contributions - overlap_count + ones_count *
        efficient_zero_contributions // (n - s + 1)
```

Figure 23: Example of a priority function found using prompt 3 and GPT-4o mini that achieves the largest $VT_0(n)$ code sizes for all lengths $n \in [6, 25]$, with 100% sequence overlap.

```
def f(v, G, n, s):
    num_ones = v.count('1')
    num_zeros = n - num_ones
    total_neighbors = len(list(G.neighbors(v)))
    balance = abs(num_ones - num_zeros) / n
    pattern_score = sum((v[i:i+b].count('1')) for b in range(1, n - s + 1) for i in range(n - b + 1))
    uniqueness_score = len(set(v)) / n
    redundancy_score = total_neighbors / (n + 1e-6)
    density = num_ones / n
    return (num_ones * redundancy_score + pattern_score + uniqueness_score - density - balance)
```

Figure 24: Example of a priority function found using prompt 4 and GPT-4o mini that achieves the largest $VT_0(n)$ code sizes for all lengths $n \in [6, 13]$, with 100% sequence overlap for even $n$ and 0% overlap for odd $n$.

counts are weighted or combined. An example is shown in Figure 24. All functions achieve the largest $VT_0(n)$ code sizes for lengths $n \in [6, 13]$, with 100% sequence overlap for even $n$ and 0% overlap for odd $n$.

```
"""
Finds large independent set in graph G where vertices are binary strings of length n.
Vertices in G are connected if they share a subsequence of length at least n - s.

Improve f_1 over its previous versions below.
Keep the code short and comment for easy understanding.
"""
import numpy as np
import networkx as nx
import itertools

def generate_graph(n, s):
    G = nx.Graph()
    sequences = [''.join(seq) for seq in itertools.product('01', repeat=n)]
    for seq in sequences:
        G.add_node(seq)
    for i in range(len(sequences)):
        for j in range(i + 1, len(sequences)):
            if has_common_subsequence(sequences[i], sequences[j], n, s):
                G.add_edge(sequences[i], sequences[j])
    return G

def has_common_subsequence(seq1, seq2, n, s):
    threshold = n - s
    if threshold <= 0:
        return True
    prev = [0] * (n + 1)
    current = [0] * (n + 1)
    for i in range(1, n + 1):
        for j in range(1, n + 1):
            if seq1[i - 1] == seq2[j - 1]:
                current[j] = prev[j - 1] + 1
            else:
                current[j] = max(prev[j], current[j - 1])
            if current[j] >= threshold:
                return True
        prev, current = current, prev
    return False

def evaluate(params):
    n, s = params
    independent_set = solve(n, s)
    return len(independent_set)

def solve(n, s):
    G_original = generate_graph(n, s)
    G_for_priority = G_original.copy()
    priorities = {v: f_1(v,G_for_priority, n, s) for v in G_original.nodes}
    vertices_sorted = sorted(G_original.nodes, key=lambda v: (-priorities[v], v))
    independent_set = set()
    for v in vertices_sorted:
        if v not in G_original:
            continue
        independent_set.add(v)
        neighbors = list(G_original.neighbors(v))
        G_original.remove_node(v)
        G_original.remove_nodes_from(neighbors)
    return independent_set

def f_0(v, G):
    """Returns the priority with which we want to add vertex v."""
    return 0.0

def f_1(v, G):
    """Improved version of f_0"""
```

Figure 25: Prompt 2 includes the evaluation script, which provides context on how the priority function is used to construct the independent set.

# H EQUIVALENCE BETWEEN OUR DISCOVERED PRIORITY FUNCTION AND THE LARGEST VT CODE

In this section, we prove that the greedy algorithm using our priority function $f$ from Figure 4 constructs the largest $\text{VT}_0(n)$ code for any code length $n$. For a single deletion, the function assigns priority to a binary sequence $\mathbf{v} \in \{0,1\}^n$ as

$$f(\mathbf{v}) = \left\lfloor \frac{W(\mathbf{v})}{n+1} \right\rfloor \quad \text{where} \quad W(\mathbf{v}) = \sum_{i=1}^{n} (n-i+1)v_i. \tag{2}$$

The priority function $f(\mathbf{v})$ groups sequences by their remainder $r(\mathbf{v}) = W(\mathbf{v}) \bmod (n+1)$. Since $W(\mathbf{v}) \equiv -\sum_{i=1}^{n} iv_i \pmod{n+1}$, sequences with the same remainder belong to the same VT code $\text{VT}_a(n)$ where $a = n + 1 - r(\mathbf{v})$, and $\mathbf{v} \in \text{VT}_0(n)$ if and only if $r(\mathbf{v}) = 0$.

To show that our greedy algorithm with priority function $f$ constructs the largest $\text{VT}_0(n)$ code, we prove that any sequence $\mathbf{w} \notin \text{VT}_0(n)$ shares a common subsequence with some sequence $\mathbf{v} \in \text{VT}_0(n)$ that has higher priority or is lexicographically smaller at the same priority.

**Lemma 1.** *For any binary sequence $\mathbf{w} \notin \text{VT}_0(n)$ with priority $f(\mathbf{w}) = q$, there exists a sequence $\mathbf{v} \in \text{VT}_0(n)$ that shares a common subsequence of length $n-1$ with $\mathbf{w}$ and satisfies one of the following conditions:*

> *1. $f(\mathbf{v}) = q+1$, or*
>
> *2. $f(\mathbf{v}) = q$ and $\mathbf{v} <_{lex} \mathbf{w}$, where $<_{lex}$ denotes lexicographically smaller.*

*Proof.* By the maximality of VT codes (Cullina et al., 2012), for any sequence $\mathbf{w} \notin \text{VT}_0(n)$, there exists at least one sequence $\mathbf{v} \in \text{VT}_0(n)$ that shares a common subsequence of length $n-1$ with $\mathbf{w}$. We show that any such $\mathbf{v}$ satisfies either condition 1 or condition 2 of Lemma 1.

Let $\mathbf{z}$ denote their common subsequence. Let $I_b^j(\mathbf{z})$ denote inserting bit $b \in \{0,1\}$ at position $j$ in $\mathbf{z}$. Then $\mathbf{v} = I_{b_v}^{j_v}(\mathbf{z})$ and $\mathbf{w} = I_{b_w}^{j_w}(\mathbf{z})$ for some bits $b_v, b_w$ and position $j_v, j_w$.

**Claim 1.** $f(\mathbf{v}) \in \{q, q+1\}$.

*Proof of Claim 1.* We have $W(\mathbf{v}) = (n+1)f(\mathbf{v})$ and $W(\mathbf{w}) = (n+1)q + r(\mathbf{w})$ where $1 \leq r(\mathbf{w}) \leq n$. Therefore

$$W(\mathbf{v}) - W(\mathbf{w}) = (n+1)(f(\mathbf{v}) - q) - r(\mathbf{w}).$$

To constrain $f(\mathbf{v}) - q$, we first bound $|W(\mathbf{v}) - W(\mathbf{w})|$. Let $m \leq n-1$ denote the number of 1-bits in the common subsequence $\mathbf{z}$.

**Case A:** $b_v = b_w = 0$. Inserting a 0-bit at different positions shifts at most $m$ ones by one position, giving $1 \leq |W(\mathbf{v}) - W(\mathbf{w})| \leq m \leq n-1$.

**Case B:** $b_v = b_w = 1$. Inserting a 1-bit at any position adds between $m+1$ (when inserted at the end) and $n$ (when inserted at the beginning) to the weight $1 \leq W(\mathbf{z})$, giving $|W(\mathbf{v}) - W(\mathbf{w})| \leq n - (m+1) \leq n-1$.

**Case C:** $b_v \neq b_w$. Without loss of generality, assume $b_v = 1$ and $b_w = 0$. Removing the inserted bits from both sequences recovers $\mathbf{z}$, giving $W(\mathbf{v}) = W(\mathbf{z}) + (n - j_v + 1)$ and $W(\mathbf{z}) - m \leq W(\mathbf{w}) \leq W(\mathbf{z})$. Thus $1 \leq W(\mathbf{v}) - W(\mathbf{w}) \leq (n - j_v + 1) + m \leq n + (n-1) = 2n - 1$. Since $j_v \geq 1$ and $m \leq n-1$, we get $1 \leq W(\mathbf{v}) - W(\mathbf{w}) \leq n$.

From Cases A-C, we have $|W(\mathbf{v}) - W(\mathbf{w})| \leq n$, which gives $|(n+1)(f(\mathbf{v}) - q) - r(\mathbf{w})| \leq n$.

If $|f(\mathbf{v}) - q| \geq 2$ or $f(\mathbf{v}) - q = -1$, then $|(n+1)(f(\mathbf{v}) - q) - r(\mathbf{w})| \geq n+2$, contradicting the bound. Therefore $f(\mathbf{v}) - q \in \{0,1\}$, as claimed. $\square$

**Claim 2.** If $f(\mathbf{v}) = q$, then $\mathbf{v} <_{\text{lex}} \mathbf{w}$.

*Proof of Claim 2.* Suppose $f(\mathbf{v}) = f(\mathbf{w}) = q$. We have $W(\mathbf{v}) = (n + 1)q$ and $W(\mathbf{w}) = (n + 1)q + r(\mathbf{w})$ where $r(\mathbf{w}) \geq 1$. Therefore $W(\mathbf{w}) > W(\mathbf{v})$.

We show that if $\mathbf{w} <_{\text{lex}} \mathbf{v}$, then $W(\mathbf{w}) \leq W(\mathbf{v})$, which contradicts $W(\mathbf{w}) > W(\mathbf{v})$. Therefore, we must have $\mathbf{v} <_{\text{lex}} \mathbf{w}$.

**Case A:** $b_v = b_w = 0$ with $j_w < j_v$ (so $\mathbf{w} <_{\text{lex}} \mathbf{v}$). The bits in $\mathbf{z}$ at positions $i < j_w$ have the same weight contribution in both sequences. For positions $i \in [j_w, j_v)$, bits shift in $\mathbf{w}$ but not in $\mathbf{v}$, contributing weight $(n - i)$ in $\mathbf{w}$ versus $(n - i + 1)$ in $\mathbf{v}$, giving $W(\mathbf{v}) - W(\mathbf{w}) = \sum_{i=j_w}^{j_v-1} z_i \geq 0$. Thus $W(\mathbf{v}) \geq W(\mathbf{w})$, which contradicts $W(\mathbf{w}) > W(\mathbf{v})$.

**Case B:** $b_v = b_w = 1$ with $j_w > j_v$ (so $\mathbf{w} <_{\text{lex}} \mathbf{v}$). The 1-bit at position $j_v$ contributes $(n - j_v + 1)$ to $W(\mathbf{v})$, and the 1-bit at position $j_w$ contributes $(n - j_w + 1)$ to $W(\mathbf{w})$. Since $j_w > j_v$, we have $(n - j_v + 1) > (n - j_w + 1)$, giving $W(\mathbf{v}) > W(\mathbf{w})$, which contradicts $W(\mathbf{w}) > W(\mathbf{v})$.

**Case C:** $b_v = 1$ and $b_w = 0$. From Case C in Claim 1, we have $W(\mathbf{v}) \geq W(\mathbf{w}) + 1 > W(\mathbf{w})$, which contradicts $W(\mathbf{w}) > W(\mathbf{v})$ an if $b_v = 0$ and $b_w = 1$, then $\mathbf{w}$ cannot be lexicographically smaller than $\mathbf{v}$.

Therefore, if $f(\mathbf{v}) = q$, then $\mathbf{v} <_{\text{lex}} \mathbf{w}$, proving the claim. $\qquad\square$

By Claim 1, any sequence $\mathbf{v} \in \text{VT}_0(n)$ that shares a common subsequence with $\mathbf{w}$ has priority $f(\mathbf{v}) \in \{q, q + 1\}$. By Claim 2, if $f(\mathbf{v}) = q$, then $\mathbf{v} <_{\text{lex}} \mathbf{w}$. It follows that $\mathbf{v}$ satisfies either condition 1 or condition 2 of Lemma 1. $\qquad\square$

Lemma 1 establishes that the greedy algorithm with priority function $f$ constructs the largest $\text{VT}_0(n)$ code for any length $n$. Any sequence $\mathbf{w} \notin \text{VT}_0(n)$ shares a subsequence with some $\mathbf{v} \in \text{VT}_0(n)$ that either has higher priority and is selected earlier, or has the same priority but is lexicographically smaller and thus selected earlier. This completes our proof of equivalence.

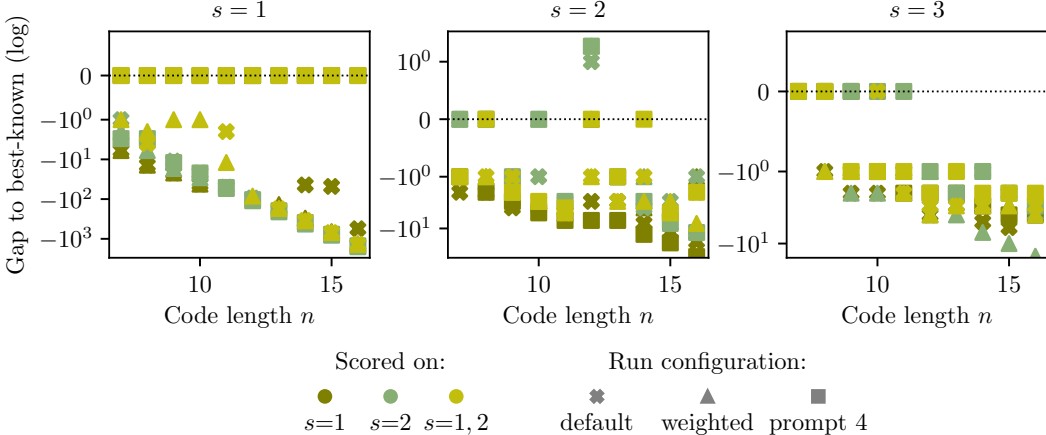

Figure 26: Gap to best-known code sizes (log scale) across all runs, varying evaluation inputs (single, two, joint deletions) and configurations (default, weighted and prompt 4).

# I DETAILS ON SEARCH FOR MULTIPLE DELETION CORRECTING CODES

In this section, we detail results from our searches for two-deletion-correcting codes, as well as joint searches for single- and two-deletion-correcting codes. We analyze both performance on evaluation inputs (i.e., the deletion parameters and code lengths used to evaluate the new functions during the search) and generalization to unseen deletion parameters and code lengths.

We consider three sets of evaluation inputs, defined by the number of deletions $s$ and the code length $n$: (i) $s = 1$, $n \in [6, 11]$; (ii) $s = 2$, $n \in [7, 12]$; and (iii) $s = 1, 2$, with $n \in [9, 11]$ for $s = 1$, and $n \in [10, 12]$ for $s = 2$. For each set, we report results using the default configuration, weighted scoring, and prompt 4.

Table 4 summarizes the code sizes achieved for single, two, and three deletions across lengths $n \in [6, 16]$. For two deletions, the search finds priority functions that match or nearly match the best-known code sizes across all tested lengths. For $n = 12$, it discovers a function (Figure 28) that constructs a code of size 34, improving upon the previous best of 32. For $n = 16$, the search for single- and two-deletion-correcting codes yields a new lower bound of 204 (e.g., achieved by the function in Figure 32), exceeding the previous best of 201.

Figure 26 shows the difference from the best-known code sizes for the functions with the smallest total difference to best-known across all deletion parameters (single, two, and three) and lengths $n \in [6, 16]$. Among all functions scored on two-deletion-correcting code sizes, the best one achieves a total difference of 2957 (normalized: 4.03). In contrast, scoring on both single- and two-deletion-correcting code sizes results in a much lower total difference of 30 (normalized: 1.75). The normalized score divides each difference by the corresponding best-known code size, ensuring that large absolute differences for single-deletion cases (where code sizes are larger) do not dominate the total. The lower scores in the joint case (both normalized and unnormalized) suggest better generalization across deletion counts and code lengths.

Table 4: Code sizes achieved for single, two, and three deletions by priority functions from runs evaluated on $s = 1$, $s = 2$, and $s = 1, 2$. Each entry is the maximum across all best-performing functions[*]. Best-performing functions are selected based on exact matches (when $s = 1$), or the smallest total difference from best-known sizes over the run's evaluation inputs (when $s > 1$). The final columns report sizes for comparison from: the trivial lexicographic baseline, prior search results (Landjev & Haralambiev, 2007), explicit known constructions ($\mathrm{VT}_0(n)$ for $s = 1$, Helberg & Ferreira (2002) for $s = 2, 3$), and best-known code sizes ($\mathrm{VT}_0(n)$ for $s = 1$, minimum-degree heuristics (Khajouei et al., 2011) for $s = 2, 3$). Bold values indicate known maxima. Superscripts link to figures showing the function that achieves the reported code size.

| $(n, s)$ | Scored on $s = 1$[**] | Scored on $s = 2$ | Scored on $s = 1, 2$ | Trivial | Search-based | Construction-based | Best known |
|---|---|---|---|---|---|---|---|
| (7,1) | 16 | 15 | 16[33] | 14 | - | 16 | **16** |
| (8,1) | 30 | 27 | 30[33] | 25 | - | 30 | **30** |
| (9,1) | 52 | 44 | 52[33] | 42 | - | 52 | **52** |
| (10,1) | 94 | 80 | 94[33] | 71 | - | 94 | **94** |
| (11,1) | 172 | 131 | 172[33] | 125 | - | 172 | **172** |
| (12,1) | 316[4,19,23] | 227 | 316[33] | 224 | - | 316 | **316** |
| (13,1) | 586[4,19,23] | 409 | 586[33] | 406 | - | 586 | **586** |
| (14,1) | 1096[4,19,23] | 743 | 1096[33] | 737 | - | 1096 | **1096** |
| (15,1) | 2048[4,19,23] | 1342 | 2048[33] | 1345 | - | 2048 | **2048** |
| (16,1) | 3856[4,19,23] | 2467 | 3856[33] | 2468 | - | 3856 | **3856** |
| (7,2) | 5[9,19] | 5[27] | 5[29] | 5 | 5 | 4 | **5** |
| (8,2) | 7[9,15] | 7[27] | 7[33] | 6 | 7 | 5 | **7** |
| (9,2) | 9 | 10 | 10 | 9 | 11 | 6 | **11** |
| (10,2) | 13 | 16[28] | 15 | 13 | 16 | 8 | **16** |
| (11,2) | 21 | 22 | 21 | 20 | 21 | 9 | **24** |
| (12,2) | 32[15] | 34[28] | 33 | 29 | 31 | 11 | 32 |
| (13,2) | 50[9] | 48 | 50[31] | 46 | 49 | 15 | 49 |
| (14,2) | 78[19] | 77 | 78[33] | 72 | 75 | 18 | 78 |
| (15,2) | 125 | 123 | 124 | 114 | 109 | - | 126 |
| (16,2) | 201[9] | 200 | 204[32] | 189 | 176 | - | 201 |
| (7,3) | 2[4,19,21,23] | 2[27,28] | 2[29] | 2 | 2 | 2 | **2** |
| (8,3) | 4[4,19] | 4[27,28] | 4[29] | 4 | 4 | 3 | **4** |
| (9,3) | 5[4] | 5[27,28] | 4 | 5 | 5 | 4 | **5** |
| (10,3) | 5 | 6[27,28] | 6[30] | 5 | 6 | 4 | 6 |
| (11,3) | 7 | 8[27,28] | 7 | 6 | 7 | 5 | 8 |
| (12,3) | 11 | 11 | 10 | 10 | 10 | 6 | 12 |
| (13,3) | 13 | 14 | 14 | 13 | 12 | 8 | 15 |
| (14,3) | 19 | 20[27] | 18 | 18 | 15 | 8 | 20 |
| (15,3) | 26 | 26 | 26 | 24 | 24 | - | 28 |
| (16,3) | 37 | 37 | 38 | 34 | 31 | - | 40 |

[*] If the maximum is taken over all priority functions in the database at the end of the search, the constructed code sizes match (or exceed, for $n = 13$) the best known sizes on all evaluation inputs.

[**] For computational reasons, we did not construct code sizes for all of the 170 successful priority functions discovered during the searches for single-deletion-correcting codes. Instead, the maximum is taken over the subset of functions shown in Figures 10, 9, 15, 19, 4, 21, 23 and 24.

```
def f(v, G, n, s):
    """Returns the priority with which we want to add `v` to the independent set."""
    nodeInt= int(v,base=2); #convert to decimal
    bitwiseXORArray= [int(i,2)^nodeInt
                        for i in list(set(G[v]))];#create array that shows what value is different between this
      and each neighbour
    numOfOnes= [(lambda x : sum(map(int, bin(x).replace('0b','')[::-1])))(bitValue)#how many ones in the
      difference
                for bitValue
                in bitwiseXORArray ];
    distBetweenBitAndNode= [(lambda x: n - abs(n // 2 - x))(onesCount) for onesCount in numOfOnes];
    avgOfDifferenceInBitsFromMedian= sum(distBetweenBitAndNode)/(max(1,(len(numOfOnes)-1)));
    score= (.9**(avgOfDifferenceInBitsFromMedian)) * ((float)(bin(nodeInt).count('1')))**(7/(1+(abs(6-n))));
    return round(score,3)
```

Figure 27: Example of a priority function found using default configuration, scored on two-deletion-correcting code sizes.

```
def f(v, G, n, s):
    """Returns the priority with which we want to add `v` to the independent set."""
    hamming_dist = [ ]
    for v in list(G.adj[v]):
        difference = [(i!= j)for (i,j) in zip(v,v )]
        dist= sum([(i ==True )for i in difference ])
        hamming_dist+= [ int(dist)]
    avg = np.array(hamming_dist).mean()
    one_count = sum([char == "1" for char in v])
    percen_one =(one_count / len(v))
    priority =.8*(avg)+ -.7* abs (((percen_one)-.5 ))
    return -round(priority,4)
```

Figure 28: Example of a priority function found using prompt 4, scored on two-deletion-correcting code sizes.

```
def f(v, G, n, s):
    """Returns the priority with which we want to add `v` to the independent set."""
    maxseqLenght= min((n*.7),(7.+s));
    kmrsLengh= max((round(np.mean ([2,maxseqLenght])) ), 3.);
    numberKmers= n-(kmrsLengh)+(1);
    kmscrLst=[]
    for stidx in range(numberKmers):
      numOfonesinNd= sum([(c=="1") *1for c in v[stidx : (stidx+(kmrsLengh))]]);
      OneWtgh= (numOfonesinNd/kmrsLengh)**.5;
      Kmrcr= (1./(OneWtgh +.000001 ))**((kmrsLengh )/2) * (numberKmers/.1)*(kmrsLengh)** -.45;
      kmscrLst.append(Kmrcr );
    Ttlscr= (np. log(((1.*numberKmers )*np. mean(kmscrLst)))).__abs__();
    return -Ttlscr
```

Figure 29: Example of a priority function found using prompt 3, scored on single- and two-deletion-correcting code sizes.

```
def f(v, G, n, s):
    """Returns the priority with which we want to add `v` to the independent set."""
    total=0
    d=[ (int(bit)) for bit in list(v)]
    degree=len(list(filter( lambda x :(int(x)==1 ),[ (int(bit)) for bit in list(v)])))
    adj = len(list(nx.neighbors(G, v)))
    if(degree<=1 and adj <7):
        return    (.9/(1.+float(degree))) *( pow((((deg+7)/2.* float(total))+0.01),(.9/.9+(1/deg)))) * pow(1./
      adj,-(.15))
    else:
        for k in range(n//2 + n %2):
            total += sum([(int)(d[i])for i in range(k,(n)-k)])
        deg=(max(degree,.1))/1.
        return ((1./(float(deg)+1))* ( (deg +1.)**deg )*total+0.01)*( pow( ( 1.-(1.-1./float(adj)) ),(-.3)))
```

Figure 30: Example of a priority function found using default configuration, scored on single- and two-deletion-correcting code sizes.

```
def f(v, G, n, s):
    """Returns the priority with which we want to add 'v' to the independent set."""
    def findNumberOfOnesForEveryPossibleSubstring():
        def numberOfOnesInNode(i,k):
            substr = v[i:(i + k)]
            return sum([int (val == '1')for val in substr]);
        possibleLengths=[x for x in range(1,(n-s))]
        onelist=[]
        for index,elemt in enumerate(possibleLengths):
            startindex= 0
            while True:
            numofOne=numberOfOnesInNode(startindex, elemt);
            onelist.append({'onenum':numofOne,'startingIndex':startindex});
            startindex += 1
                if ((startindex+ elemt)>n):
                    break
        return onelist
    onelist=findNumberOfOnesForEveryPossibleSubstring()
    score=lambda x:-(x['onenum'] * x['onenum']) *(max(1,abs(((x['startingIndex']/float(n)))-(s/(float(n)))))))
    finalScore=map(score,onelist)
    return sum(finalScore)
```

Figure 31: Example of a priority function found using prompt 3, scored on single- and two-deletion-correcting code sizes. It achieves a new lower bound for $s = 2$ and $n = 16$, with size 202, compared to the previously best known size of 201.

```
def f(v, G, n, s):
    """Returns the priority with which we want to add 'v' to the independent set."""
    weight= []
    for k in range ((n)+1):
        cnt=0
        for p in range(((n)- (k))+1):
            substring=""
            for r in range(p,(p)+(k)):
                characTer=str( int(v[r]))
                substring+=characTer
            numZeROES=substring.count("0")
            NUMONES=substring.count("1")
            if numZeROES>=NUMONES:
                Weight=-( numZeROES*2*(k+1))
            else :
                    Weight =(NUMONES*.8*(k+1))
            weight.append(Weight)
    averagE=np.mean(weight )
    return averagE
```

Figure 32: Example of a priority function found using prompt 4, scored on one and two-deletion correcting code sizes. It achieves a new lower bound for $s = 2$ and $n = 16$, with size 204 compared to the previously best known size of 201.

```
def f(v, G, n, s):
    """Returns the priority with which we want to add 'v' to the independent set."""
    wt=[]
    for q in range ((n)+1):
        counter=0
        for w in range(((n)-q )+1) :
            substring=""
            for e in range(w, (w +(q))):
                character= str( int(v[e]))
                substring+= character
            NumberofZeroes=substring.count("0")
            NumbersOfOnes=substring.count("1")
            if NumbersOfOnes>=NumberofZeroes :
                weight= -(NumbersOfOnes )*(q*6+.89)
            else :
                    weight= (NumberofZeroes )*.5 * (q *3 )
            wt.append(weight)
    if len(wt)!=0 :
        Average=sum(wt)/len(wt)**.7*3
    return Average
```

Figure 33: Example of a priority function found using prompt 4, evaluated on one and two-deletion-correcting code sizes.

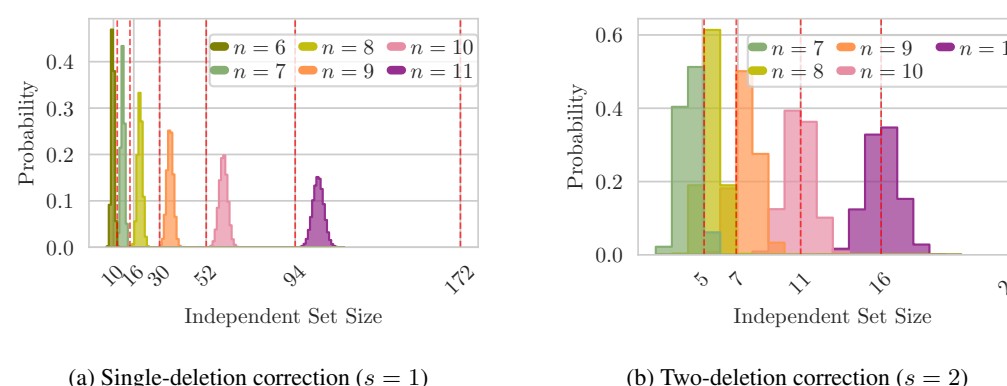

(a) Single-deletion correction ($s = 1$)  (b) Two-deletion correction ($s = 2$)

Figure 34: Distribution of independent set sizes when sequences are iteratively added in order over $10^5$ permutations of all $2^n$ sequences.

Table 5: Code sizes found by random construction, KaMIS, and our approach (maximum over three runs). Bold indicates known maximum sizes.

| | | | | | | Single-deletion correction ($s = 1$) | | | | | | | | | |
|---|---|---|---|---|---|---|---|---|---|---|---|---|---|---|---|
| Algorithm | $n = 6$ | 7 | 8 | 9 | 10 | 11 | 12 | 13 | 14 | 15 | 16 | 17 | 18 | 19 | 20 |
| Random | **10** | **16** | 26 | 42 | 69 | 116 | 201 | 351 | 620 | 1111 | 2003 | 3636 | 6644 | 12215 | - |
| OnlineMIS | **10** | **16** | **30** | **52** | **94** | 143 | 254 | 456 | 816 | 1469 | 2657 | 4843 | 8908 | 16439 | 30440 |
| ReduMIS | **10**[†] | **16**[†] | **30**[†] | **52**[†] | **94**[†] | 172[†] | 316[†] | 457[†] | 812[†] | 1453[†] | 2634[†] | 4803[†] | 8793[†] | 16180[†] | 30111 |
| Our approach | **10** | **16** | **30** | **52** | **94** | **172** | 316 | 586 | 1096 | 2048 | 3856 | 7286 | 13798 | 26216 | 49940 |

([†]) Solver stopped due to its no-improvement stopping rule, not the 24-hour time limit.

## J  PROBLEM DIFFICULTY AND BASELINES

In this section, we analyze problem difficulty and compare our discovered code sizes against random codes and the state-of-the-art general maximum independent set solver KaMIS (Andrade et al., 2012; Lamm et al., 2016). We do not evaluate neural solvers, as they currently underperform classical methods on standard graph benchmarks (Böther et al., 2022; Wu et al., 2025).

Finding a maximum independent set in a general graph is NP-complete (Lovász & Plummer, 2009). If we know the maximum code size $C^*$, we must evaluate all $\binom{2^n}{C^*}$ subsets of that size. For $n = 6$ (the smallest length we consider), $s = 1$, and $C^* = 10$, this already exceeds 151 billion evaluations. If we do not know the maximum size, we must consider all possible subset sizes, which leads to a worst-case complexity of $2^{2^n}$. Verifying whether a subset forms a valid code requires checking that no two sequences share a common subsequence of length $n - s$ (or equivalently, are within edit distance $2s + 1$ for codes correcting up to $s$ insertions, deletions, and substitutions). These checks use dynamic programming with $O(n^2)$ time per pair for fixed $s$, so verifying a subset of size $k$ requires $O(k^2 n^2)$ time (Cormen et al., 2022), giving total worst-case complexity $O(2^{2^n} \cdot k^2 n^2)$.

However, when graphs contain many maximum independent sets, simple random construction can find one quickly. To test this, we sample $10^5$ random orderings of all $2^n$ sequences (across three runs with different random seeds) and count how often greedy construction finds a maximum independent set for $s = 1$ ($n \in [6, 11]$) and $s = 2$ ($n \in [7, 12]$). Figure 34 shows the distribution of code sizes. For $n = 6$ ($s = 1$) and $n = 7$ ($s = 2$), random construction succeeds in $119.7 \pm 9$ and $6146.3 \pm 82.2$ of $10^5$ attempts, respectively. For $n = 7$ ($s = 1$) and $n = 8$ ($s = 2$), it succeeds in $6.0 \pm 1.0$ and $297.7 \pm 12.9$ attempts. For larger $n$, it does not find maximum-size sets.

We further compare our LLM-guided approach against two algorithms from the KaMIS framework, OnlineMIS (Andrade et al., 2012) and ReduMIS (Lamm et al., 2016). We run both algorithms with three different seeds on our graphs for lengths $n \in [6, 20]$ and a single deletion $s = 1$, using a 24-hour timeout. Table 5 reports the maximum code size found across the three runs for each algorithm. KaMIS achieves the best known sizes for $n \leq 12$. For larger lengths, it finds smaller codes, with the gap increasing as $n$ increases.

