# OpenReview forum: "LLM-Guided Search for Deletion-Correcting Codes"
_ICLR.cc/2026/Conference — Submitted to ICLR 2026_

### Official Review · Reviewer_72eE · 2025-10-28

**Soundness:** 2
**Presentation:** 3
**Contribution:** 2
**Rating:** 2
**Confidence:** 4

**Summary:**

The paper adapts FunSearch to construct deletion‑correcting codes. The core idea is to search for a priority function that greedily builds a large independent set in a graph whose vertices are binary strings and edges connect strings sharing a length $n−s$ subsequence. The authors add a deduplication step (hashing the per‑sequence priority vector) to avoid storing functionally identical candidates. Empirically, they (i) match known optima for single deletion up to $n\leq 11$ and reach $VT_0$ sizes for larger $n$; (ii) report new lower bounds for two deletions at $n=12,13,16$.

**Strengths:**

Applying LLM (FunSearch) to coding theory. Casting single/multiple deletion code construction as independent‑set search with an LLM‑discovered priority function is a clean mapping and is described systematically.

**Weaknesses:**

1. The method is essentially *vanilla FunSearch* plus a simple deduplication based on hashing the priority scores a function assigns to the evaluation inputs. While *Fig. 12* shows 20 % duplicates without the filter and a faster trajectory with dedup, there’s no head-to-head comparison against recent FunSearch variants under matched resources to establish a genuine sample-efficiency gain. The contribution feels primarily engineering rather than algorithmic.

2. Practical impact on coding theory is small.  For $s = 1$, the system mostly re-discovers $VT_0$ (or achieves the same sizes); the truly new outcomes are modest lower-bound improvements for $s = 2$ at a few lengths—$n = 12 : 32 \rightarrow 34$, $n = 13 : 49  \rightarrow 50$, $n = 16 : 201 \rightarrow 204$. Given the stated budget (~400 K functions per run; 350 GPU hours), the improvement-per-compute looks low.

3.  On the combinatorial side: no comparisons to ILP / SAT / CP-SAT or LP-relaxation + heuristics that are common for maximum-independent-set / coding-theory graphs at these sizes. On the LLM-guided side: no matched comparisons to ReEvo / EoH style search with the same evaluator and budget. The only ablations are within-prompt / model variants, which do not establish that the framework is stronger than alternatives.

**Questions:**

1. Can you provide a fully formal argument for Appx. H, especially the conditions under which lexicographic tie‑breaking ensures that remainder $r=0$ sequences are always exhausted first at a fixed quotient? and discuss whether the discovered forms say anything structural for $s>1$.

2. Please compare to graph/LP‑based solvers on the same $n,s$, with wall‑clock and solution quality, to justify LLM‑guided search as the preferred tool.

---

> ### Author Response · Authors · 2025-11-25
>
> We thank the reviewer for their careful reading and constructive feedback. We appreciate the concerns raised and address each one below.
>
> **Response to Weakness 1**
>
> We agree that the deduplication step is primarily an engineering contribution that addresses a practical issue we observed, where the LLM frequently generated syntactically different but functionally identical priority functions, leading to expensive re-evaluations and reduced exploration. Our hash-based approach, comparing the priority scores assigned to sequences, was effective for our problem and is simpler than the embedding-based methods for determining functional equivalence that appear in follow-up work on FunSearch, such as [1].
>
> Regarding comparisons to ReEvo/EoH, our goal was to show that LLM-guided search is promising for coding theory, not to propose a competing search method. Our contribution is the first application of LLM-guided evolutionary search in this domain, where existing approaches rely on fundamentally different techniques. We show that LLM-guided search can (1) rediscover known optimal codes, (2) find new codes matching conjectured optimal sizes, and (3) establish new lower bounds. This demonstrates that LLM-guided search is a promising direction for code design.
>
> **Response to Weakness 2**
>
> The new lower bounds we report are limited to $n=12,13,16$ for $s=2$,  but we want to highlight the following contributions:
>
> - **Single-deletion correction:** Our search independently rediscovered conjectured-optimal $VT_0(n)$ codes in an alternative formulation (Figure 4) and also found constructions with zero sequence overlap (Figure 10). This shows that LLM-guided search can recover known optimal constructions without any explicit domain knowledge.
>
> - **Two-deletion correction:** Finding deletion-correcting codes has been studied for over 70 years. Previous computational searches searched the full $2^n$ sequence space, which becomes infeasible for moderate $n$ and does not give an interpretable representation of the solutions. Our function-space search generalizes better, as shown by functions constructing codes beyond what they were optimized for and is interpretable.
>
> - **Compute cost:** The 350 GPU hours is a one-time search cost that results in an interpretable priority function. Once discovered, this function can be applied to construct codes of any length.
>
> **Response to Weakness 3 and Question 2**
>
> Regarding comparisons to ReEvo/EoH, please see our response to Weakness 1.
> Regarding comparisons to ILP/SAT/LP-relaxation and other graph solvers. We acknowledge this limitation. Our main goal was to test whether LLM-guided search can discover interpretable priority functions for constructing deletion-correcting codes. We used the single-deletion setting as a controlled regime where optimal sizes are partly known, allowing us to validate whether the LLM can discover functions that construct codes matching these optimal sizes. Since our focus was on understanding the discovered logic and its connection to coding theory (e.g., rediscovering VT codes), we initially compared against best-known code sizes from the literature rather than general independent set solvers.
> However, we agree that direct comparisons are valuable. We have now added comparisons to two widely used maximum independent set solvers:
> - OnlineMIS [2,3]: local-search heuristic with kernelization.
> - ReduMIS [4]: iterative kernelization with evolutionary local search.
>
> We used the KaMIS implementation with a 24-hour timeout, running each solver three times with different seeds and reporting the maximum code size achieved. We did not compare to neural independent set solvers, as prior work [5,6] shows classical solvers consistently outperform current neural approaches on standard graph benchmarks. The results are below and in the revised manuscript in Appendix J.
>
> Regarding LP relaxation, prior work on upper bounds for deletion-correcting codes uses LP relaxations of the maximum independent set formulation (e.g., [7]). These methods provide upper bounds on the best possible code size and are complementary to constructive approaches. Our focus is on generating actual codes (lower bounds).

---

> > ### Author Response · Authors · 2025-11-25
> >
> > | n  | Random | OnlineMIS | ReduMIS | Our approach |
> > |----|--------|-----------|---------|-----------------------|
> > | 6  | 10     | 10        | 10*     | 10           |
> > | 7  | 16     | 16        | 16*     | 16           |
> > | 8  | 26     | 30        | 30*     | 30           |
> > | 9  | 42     | 52        | 52*     | 52           |
> > | 10 | 69     | 94        | 94*     | 94           |
> > | 11 | 116    | 143       | 172*    | 172          |
> > | 12 | 201    | 254       | 316*    | 316          |
> > | 13 | 351    | 456       | 457*    | 586          |
> > | 14 | 620    | 816       | 812*    | 1096         |
> > | 15 | 1111   | 1469      | 1453*   | 2048         |
> > | 16 | 2003   | 2657      | 2634*   | 3856         |
> > | 17 | 3636   | 4843      | 4803*   | 7286         |
> > | 18 | 6644   | 8908      | 8793*   | 13798        |
> > | 19 | 12215  | 16439     | 16180*  | 26216        |
> > | 20 | –      | 30440     | 30111   | 49940        |
> > *Solver stopped due to its no-improvement stopping rule, not the 24-hour limit.

---

> ### Author Response · Authors · 2025-11-25
>
> **Response to Question 1**
>
> We have restructured the proof in Appendix H to make it more clear and formalized. The proof now consists of one main lemma and two claims that establish it:
>
> - Lemma: For any sequence $\mathbf{w} \notin \mathrm{VT}_0(n)$, there exists a conflicting sequence $\mathbf{v} \in \mathrm{VT}_0(n)$ with either higher priority or the same priority but lexicographically smaller.
> - Claim 1 establishes that any such $\mathbf{v}$ has priority $f(\mathbf{v})=q$ or $q+1$, where $q = f(\mathbf{w})$.
> - Claim 2 proves that if $f(\mathbf{v}) = q$, then $\mathbf{v} <_{\text{lex}} \mathbf{w}$.
>
> We prove both claims through case analysis on how sequences can differ from their common subsequence. Claim 2 makes explicit the conditions under which lexicographic tie-breaking ensures that sequences with remainder $r(\mathbf{v}) = 0$ are selected before others at the same priority.
>
> **Regarding structural insights from the discovered formulation:** We could not identify substantial structural implications beyond equivalence to the standard VT codes. An implementation consideration for LLM-guided search we found is that specifying the tie-breaking rule is important for the performance of discovered priority functions. Our original implementation used an unspecified tie-breaking rule (“take the first sequence”) which was sensitive to graph construction order. When we constructed graphs differently, priority functions that were optimal under the original construction performed poorly.
>
> We hope this addresses the point raised and are happy to discuss further if needed.
>
> [1] Lange, Robert Tjarko, Yuki Imajuku, and Edoardo Cetin. "Shinkaevolve: Towards open-ended and sample-efficient program evolution." arXiv preprint arXiv:2509.19349 (2025).
>
> [2] Andrade, Diogo V., Mauricio GC Resende, and Renato F. Werneck. "Fast local search for the maximum independent set problem." Journal of Heuristics (2012).
>
> [3] Dahlum, Jakob, et al. "Accelerating local search for the maximum independent set problem." International symposium on experimental algorithms (2016).
>
> [4] Lamm, Sebastian, et al. "Finding near-optimal independent sets at scale." 2016 Proceedings of the eighteenth workshop on algorithm engineering and experiments (ALENEX). Society for Industrial and Applied Mathematics (2016).
>
> [5] Böther, Maximilian, et al. "What's wrong with deep learning in tree search for combinatorial optimization." arXiv preprint arXiv:2201.10494 (2022).
>
> [6] Wu, Y., Zhao, H., & Arora, S. "Time to Rethink AI for Combinatorial Optimization: Classical Algorithms Remain Tough to Match." arXiv preprint arXiv:2502.03669 (2025).
>
> [7] Cullina, Daniel, and Negar Kiyavash. "Generalized sphere-packing bounds on the size of codes for combinatorial channels." IEEE Transactions on Information Theory (2016).

---

### Official Review · Reviewer_uCM2 · 2025-10-31

**Soundness:** 2
**Presentation:** 2
**Contribution:** 3
**Rating:** 6
**Confidence:** 2

**Summary:**

The paper studies the classical problem of constructing error-correcting codes that can correct a fixed number $s$ of deletions in binary sequences of length $n$. The proposed method builds on the existing \textit{FunSearch} framework, which uses an LLM-guided evolutionary search to identify effective priority functions. The authors further introduce a deduplication step to remove functions that differ only syntactically. Using this approach, the method discovers priority functions that yield codes with new best-known sizes for code lengths $n = 12, 13,$ and $16$, thereby improving existing lower bounds.

**Strengths:**

1. Combining an LLM with an evolutionary search to design priority functions for greedy code construction is an interesting and promising direction that links learning-based methods with classical coding theory.

2. The empirical results demonstrate that the proposed method identifies codes with new best-known sizes for code lengths $n = 12, 13,$ and $16$, thereby improving the existing lower bounds. This provides supporting evidence for the effectiveness of the approach.

**Weaknesses:**

1. A large portion of the proposed method appears to rely on the existing FunSearch framework, making it unclear what specific technical contributions or novel insights are introduced in this work.
2. The approach is computationally expensive --- the paper reports approximately 350 GPU hours for processing around 400k functions for small $n$. For larger $n$, the evaluator becomes infeasible due to the exponential growth of the sequence space, which significantly limits the method’s practical applicability. Although the authors acknowledge this issue, the limitation remains substantial.
3. The description of some concepts in this paper, such as the islands/clustering logic and priority function representation, could be clearer.

**Questions:**

1. The LLM occasionally generates invalid or non-executable code. How frequently does this issue occur, and what specific strategies are used to detect and handle such cases?
2. The paper introduces a deduplication step to promote diversity among generated functions. However, this step appears to be relatively simple. Could the authors clarify the technical insights or contributions associated with this component?

---

> ### Author Response · Authors · 2025-11-25
>
> We thank the reviewer for their constructive feedback, and for recognizing the potential of combining LLM-guided search with classical coding theory as well as our empirical improvements to best-known lower bounds. Below we respond to all concerns and questions raised.
>
> **Response to Weakness 1**
>
> While our work uses the FunSearch pipeline as a general framework for LLM-guided evolutionary search, we make the following contributions.
>
> - Regarding novel insights: We are the first to apply LLM-guided evolutionary search to error-correcting code construction, where traditional approaches rely on algebraic constructions and exhaustive search rather than learned heuristics. This serves as a proof-of-concept that LLM-guided search is promising for coding and information theory and may inspire rethinking how other problems in the field, such as encoder/decoder design, can be reformulated to leverage frameworks like FunSearch.
>
> - Regarding technical contributions:  We provide an asynchronous distributed implementation with independent workers for database operations, LLM sampling, and evaluation communicating via message passing. Each worker operates asynchronously (e.g., samplers fetch new prompts while awaiting LLM responses, evaluators load graphs while constructing codes for other functions). The system supports multi-node execution with dynamic addition of workers during runtime and automatic scaling based on workload. This contrasts with existing implementations we are aware of (e.g., OpenEvolve, EoH or ReEvo), which use a single-node worker pool where each worker performs full iterations (prompt construction, LLM sampling and evaluation) sequentially with non-batched LLM inference, distributing total iterations equally across workers. Also at the time of our research, no open-source method was available.
>
> **Response to Weakness 2**
>
> While we acknowledge the computational cost and exponential scaling, we view this work as a proof-of-concept demonstrating that LLM-guided search is promising for error-correcting code discovery
>
> Regarding overcoming the scalability issue, we believe a promising future direction is searching for encoding functions directly instead of functions that construct the codebook directly. This approach would allow evaluation via Monte Carlo simulation over sampled messages rather than exhaustive sequence enumeration, avoiding the exponential scaling. However, this requires a parameterization of encoding functions that allows searching over them, which is non-trivial.
>
> **Response to Weakness 3**
>
> We thank the reviewer for this feedback. In response, we have revised Section 4 (Method) to improve clarity regarding the islands/clustering logic and priority functions. We hope these changes improve clarity, and we are happy to make further changes if needed.
>
> **Response to Question 1**
>
> Non-executable code occurs frequently, StarCoder2-15B generates executable functions 16.2% of the time and GPT-4o mini 43.7%. We simply discard non-executable functions (due to syntax errors, runtime exceptions, or 5-minute timeout). Alternatively, we could prompt the LLM again with error feedback, but we believe this limitation will be naturally addressed by using more capable coding LLMs, e.g., GPT-4o mini shows a much higher executable rate compared to StarCoder2-15B.
>
> **Response to Question 2**
>
> We agree that the deduplication step is relatively simple and primarily an engineering contribution. During our initial experiments, we observed that the search frequently became stuck in local optima, with the LLM repeatedly generating syntactically different but functionally identical priority functions. This resulted in expensive re-evaluations of the same logic and reduced exploration. The deduplication step was a simple fix we implemented to address this issue, and we observed that it successfully resolved the problem.
>
> Follow-up work on FunSearch (e.g., [1]) use embedding models to measure functional similarity between generated programs and remove very similar functions. For our setting, the simpler approach of hashing the priority scores assigned to sequences, i.e. comparing similarity at the output level rather than at the function level, was sufficient.
>
> We hope this clarifies the reviewer's concern and are happy to provide additional details if needed.
>
> [1] Lange, Robert Tjarko, Yuki Imajuku, and Edoardo Cetin. "Shinkaevolve: Towards open-ended and sample-efficient program evolution." arXiv preprint arXiv:2509.19349 (2025).

---

### Official Review · Reviewer_YB4w · 2025-11-01

**Soundness:** 3
**Presentation:** 3
**Contribution:** 3
**Rating:** 2
**Confidence:** 4

**Summary:**

This paper introduces an LLM-guided evolutionary search approach for constructing deletion-correcting codes—an open combinatorial problem in information theory. Building upon FunSearch (Romera-Paredes et al., 2024), the authors adapt the framework to generate and iteratively improve priority functions that define greedy algorithms for building large deletion-correcting codes. Using models like StarCoder2 and GPT-4o mini, they rediscover known optimal codes (Varshamov–Tenengolts, VT) and find new best-known lower bounds for two-deletion cases. The method effectively searches function space rather than code space, demonstrating LLMs’ potential in theoretical code design.

**Strengths:**

1. First known use of LLM-guided program synthesis for designing error-correcting codes—a fresh bridge between information theory and AI-driven heuristic search.
2. Successfully rediscovered VT codes and found new best-known constructions for small two-deletion cases (n = 12, 13, 16).
3. The deduplication step for function-space search improves efficiency and diversity, addressing a key limitation of prior FunSearch implementations.
4. Code release and transparent description of distributed implementation (RabbitMQ-based) facilitate future research.
5. The idea extends to other types of combinatorial or coding-theory problems beyond deletions.

**Weaknesses:**

1. Evaluation cost grows exponentially with sequence length n, making the approach infeasible for medium-to-large codes. Usually in practice long codes are required, so this algorithm cannot be applied.
2. Success heavily depends on hand-crafted prompts and model choice; reproducibility across models and random seeds may vary.
3. Experiments mostly cover short sequences (n ≤ 25); performance for longer lengths or higher deletion counts is unexplored.
4. Please compare to the existing methods for deletion channels.

**Questions:**

1. How sensitive are the discovered priority functions to LLM architecture or sampling temperature?
2. Can the approach be extended to insertion- or substitution-correcting codes with minimal modification?
3. Is there a way to approximate evaluation (e.g., sampling or probabilistic scoring) to mitigate exponential scaling?

---

> ### Author Response · Authors · 2025-11-25
>
> We thank the reviewer for their review and for recognizing the novelty of applying LLM-guided search to error-correcting code design. We appreciate the constructive feedback and address all weaknesses and questions below.
>
> **Response to Weakness 1:**
>
> We agree that our approach currently does not scale well to large code lengths, a limitation we address in Section 6 of our paper. However, finite-length deletion-correcting codes are also practically relevant:
>
> - **Short codes for indexing and barcoding:** For example, in DNA data storage, current synthesis technology does not support long DNA strands, so information must be broken into pieces and stored in unordered pools. These sequences require short indices that are distinguishable under errors to allow correct reordering. The index in [1], for example, uses sequences of length 15 nucleotides. Other applications include random access in DNA data storage (e.g., [2] uses file identifiers of length 25) and sample identification in biological sequencing to distinguish which sample a DNA sequence came from (e.g., [3] use indices of length 6).
> - **Segment-based error-correcting codes for longer codes:** Many existing deletion-correcting codes for larger code lengths rely on inserting known markers at fixed positions to break long sequences into shorter segments, then applying short deletion-correcting codes (such as VT codes) to each segment to handle multiple deletions [4,5]. Therefore, progress on short code constructions directly translates to better code rates for these marker-based schemes.
>
> **Response to Weakness 2 and Question 1:**
>
> We acknowledge that performance can vary across runs due to the stochastic nature of LLM-guided search. Appendix F shows that under identical settings (T = 0.1, P = 30K, R = 1.2K), two out of three runs discover functions that construct maximum-size codes where known.
>
> Regarding model choice, we test both a code-specialized model (StarCoder2-15B) and a general-purpose instruction-tuned model (GPT-4o mini) in Section 5.4. Both models successfully discover priority functions that construct maximum-size codes, though GPT-4o mini requires prompt engineering while StarCoder2 succeeds with the baseline prompt.
>
> We also ran a sweep over the LLM sampling temperature (Appendix B). Lower temperatures reduce diversity and increase the number of executable functions, but they do not improve sample efficiency. We then tried decreasing the temperature dynamically during the search to shift from exploration to exploitation. This generated more executable functions but also did not improve sample efficiency over a fixed temperature (Appendix C, Table 2d).
>
> Additionally, we want to highlight that for discovery tasks such as code design, the goal is to find good solutions once rather than ensure reproducibility across all runs. Once a priority function or code is discovered, it becomes a reusable contribution that does not need to be rediscovered.
>
> **Response to Weakness 3:**
>
> While our experiments focus on short sequences (n ≤ 25) and small deletion counts (s ≤ 3), our approach has advantages over prior search-based methods that directly enumerate sequences.
>
> First, the priority functions we discover are mathematically interpretable and can be analyzed to derive code sizes for arbitrary lengths without explicit construction. For example, we demonstrate this in Appendix H, where we prove that one of our discovered priority functions (Figure 4) is mathematically equivalent to the $VT_0(n)$ codes. This equivalence allows us to determine code sizes for any length n through analysis rather than construction.
>
> Second, our priority functions generalize across code lengths and deletion parameters, as shown in Section 5.3 and Table 1 and 4. Functions optimized for single deletion and $n \in [6,11]$ construct codes matching $VT_0(n)$ sizes (verified up to n = 25) and achieve competitive sizes for two and three deletions without rerunning the search. In contrast, prior search-based approaches [8, 9] search the space of $2^n$ binary sequences directly, resulting in a fixed codebook for each specific (n, s) pair without generalization.

---

> ### Author Response · Authors · 2025-11-25
>
> **Response to Weakness 4:**
>
> To the best of our knowledge, we compare our approach to the most relevant existing construction and search methods for finite-length deletion-correcting codes in Section 2.2 (Related Work) and throughout the paper.
>
> - **Construction-based methods:** We compare against VT codes [6], which are conjectured optimal for single deletion, and extensions for multiple deletions [7]. Our results match VT code sizes where they are conjectured optimal (Table 1, verified up to n = 25). For two deletions, our approach improves on the explicit construction in [7], as shown in Table 4. We added a column with the code sizes from [7] to make the comparison clearer, since their constructions are closest to our priority functions (both approaches give a generalizable, interpretable constructions that work across code lengths).
>
> - **Search-based methods:** We compare our results to prior heuristic search approaches [8,9] that directly search the space of $2^n$ binary sequences. As discussed in Section 5.3, our approach discovers larger two-deletion-correcting codes for $n \in [12, 16]$ than these methods.
>
> - **Baseline comparison:** In Appendix J of the revised manuscript, we now additionally compare against the state-of-the-art maximum independent set solver KaMIS [10].
>
> We did not include asymptotic constructions in our comparison because methods designed to achieve good asymptotic redundancy perform poorly at the finite lengths we study, where large constant factors dominate. If the reviewer has specific constructions in mind that we may have missed, we would be happy to incorporate them into the revised manuscript.
>
>
> **Response to Question 2:**
>
> Yes, our approach extends naturally to other error types with minimal modification. The change required is in the graph construction step of the evaluation script. For example, vertices should be connected if two sequences have edit distance smaller than 2s + 1 for codes correcting s insertions, deletions, and substitutions. The LLM-guided search framework, priority function optimization, and greedy construction algorithm are unchanged.
>
> **Response to Question 3:**
>
>
> We agree that addressing the scaling issue is important. In our current formulation, sampling-based approximation is challenging because we evaluate priority functions by the size of the codes they construct, so we need to actually build the full codebook to compute the evaluation score.
>
> We could consider approximation strategies such as graph partitioning, where we partition the $2^n$ sequences by structural properties (e.g., Hamming weight) and evaluate priority functions on each partition separately. However, this still requires building the full graph, assigning priorities to all $2^n$ sequences, and constructing the complete codebook. It may provide some memory or parallelization benefits but does not overcome the fundamental exponential cost. Similarly, multi-stage cascade evaluation (as in AlphaEvolve) could filter candidates on short lengths before evaluating on longer ones, reducing the number of expensive evaluations, but each evaluation still has exponential cost. Both approaches might extend our method from to perhaps lengths 25-30, but don't fundamentally change the scaling.
>
> To really overcome exponential scaling and make LLM-guided search feasible for lengths > 100, we believe a promising direction is instead of searching for construction algorithms, searching for explicit encoding functions. We could then evaluate candidates via Monte Carlo simulation by sampling random messages, encoding them, applying deletions, attempting decoding, and measuring success rate. This would eliminate the need to construct full codebooks, with the tradeoff that it guarantees low probability of decoding error rather than the zero-error guarantee. However, this requires a parameterization of encoding functions that makes them suitable to search over with this framework, which is non-trivial.
>
>
> We hope this addresses the reviewer’s concern and are happy to clarify any remaining points if needed.

---

> ### Author Response · Authors · 2025-11-25
>
> [1] Goldman, Nick, et al. "Towards practical, high-capacity, low-maintenance information storage in synthesized DNA." Nature (2013).
>
> [2]Organick, Lee, et al. "Random access in large-scale DNA data storage." Nature biotechnology (2018).
>
> [3] Craig, David W., et al. "Identification of genetic variants using bar-coded multiplexed sequencing." Nature methods (2008).
>
> [4] Mahed Abroshan, Ramji Venkataramanan, and Albert Guill´en i F`abregas. Coding for segmented
> edit channels. IEEE Transactions on Information Theory (2017).
>
> [5]  Zhenming Liu and Michael Mitzenmacher. Codes for deletion and insertion channels with segmented errors. IEEE Transactions on Information Theory (2009).
>
> [6] R. R. Varshamov and G. M. Tenengolts. Codes which correct single asymmetric errors. Avtomatika
> i Telemekhanika (1965).
>
> [7] A.S.J. Helberg and H.C. Ferreira. On multiple insertion/deletion correcting codes. IEEE Transac-
> tions on Information Theory (2002).
>
> [8] Swart, Theo G., and Hendrik C. Ferreira. "A note on double insertion/deletion correcting codes." IEEE Transactions on Information Theory (2003).
>
> [9] Landjev, Ivan, and Kristiyan Haralambiev. "On multiple deletion codes." Serdica Journal of Computing (2007).
>
> [10] Sebastian Lamm, Peter Sanders, Christian Schulz, Darren Strash, and Renato F Werneck. Find-
> ing near-optimal independent sets at scale. In Proceedings of the eighteenth workshop on
> algorithm engineering and experiments (2016).

---

### Author Response · Authors · 2025-12-03

Dear AC,

Many thanks for handling our paper.

While reviewers acknowledge the novelty of our work, calling it "an interesting and promising direction that links learning-based methods with classical coding theory" (uCM2) and "a fresh bridge between information theory and AI-driven heuristic search" (YB4w), and recognize our empirical results including rediscovering conjectured-optimal VT codes and establishing new best-known lower bounds, we believe this is not fully reflected in the scores.

The main concerns raised were practical relevance of short codes, comparisons to existing solvers, and technical contributions beyond FunSearch. We addressed all of these in our rebuttal:
- **Practical relevance of short codes** (YB4w): We clarified that short deletion-correcting codes are practically important for DNA storage indexing and barcoding, and that progress on short codes directly improves marker-based schemes for longer sequences, which are commonly used to handle multiple deletions. We also clearly acknowledge the scaling limitation to long codes in our paper.

- **Comparisons to graph solvers** (72eE): We added comparisons to state-of-the-art maximum independent set solvers (OnlineMIS, ReduMIS). Our discovered priority functions outperform these solvers.

- **Technical contributions beyond FunSearch** (uCM2, 72eE): We are the first to leverage LLM-guided evolutionary search to error-correcting code construction, serving as a proof-of-concept that this approach is promising for coding theory. We also provide an asynchronous distributed implementation supporting multi-node execution and dynamic scaling, contrasting with existing implementations (e.g., OpenEvolve, EoH, ReEvo) that use single-node sequential workers.  At the time of our research, no open-source FunSearch implementation was available.

We believe this work makes a valuable contribution as the first demonstration that LLM-guided search can discover interpretable, generalizable constructions for error-correcting codes, providing a new approach for revisiting long-standing open problems in coding and information theory. We hope the AC will consider that the main concerns have been addressed.

---

### Meta-Review · Area_Chair_qbAQ · 2025-12-08

**Summary:**

The reviewers appear to see good potential in the paper, but also highlight quite significant limitations including scalability, doubts on the added value, and limited comparisons.  Some of these concerns were (partially) addressed, e.g., via comparing to two MIS solvers.  Overall, it becomes very subjective at what point the paper deserves to be published vs ought to be worked on more.

**Reviewer Concerns:**

- The authors acknowledged the short-length limitations.  This was one of the main recurring concerns.
- Some MILP-based baselines were added but without much detailed discussion/explanation for the better performance of the proposed approach in the empirical comparison. Hence, it is a bit difficult to assess them clearly.
- Some concerns of novelty (e.g., use of FunSearch but not seeing clear new ideas) would likely still remain.
- The proof in Appendix H is re-done but not straightforward to readily verify.
- Some concerns about logic/clarity may remain.
- The authors stated "our goal was to show that LLM-guided search is promising for coding theory", but to better understand to what extent this is true and in what generality, including results of ReEvo/EoH would still be beneficial.

**Reviewer Scores:**

I wish we could have had a proper reviewer discussion period to assess this more confidently, but without that, I have to simply use my best judgment on the most likely outcome, which is that the two reviewers recommending rejection would have continued to do so (albeit perhaps less strongly than before).  This is partly because they both indicated non-minor additional work being needed, and partly because several of the author responses weren’t really disputing the concerns themselves (but were instead arguing how the content is sufficient despite those concerns).  See above for a partial list.

---

### Decision · Program_Chairs · 2026-01-26

Reject